# The genotype-phenotype landscape of an allosteric protein

Drew S Tack[1] (iD), Peter D Tonner[1] (iD), Abe Pressman[1], Nathan D Olson[1] (iD), Sasha F Levy[2,3] (iD), Eugenia F Romantseva[1] (iD), Nina Alperovich[1], Olga Vasilyeva[1] & David Ross[1,*] (iD)

## Abstract

Allostery is a fundamental biophysical mechanism that underlies cellular sensing, signaling, and metabolism. Yet a quantitative understanding of allosteric genotype-phenotype relationships remains elusive. Here, we report the large-scale measurement of the genotype-phenotype landscape for an allosteric protein: the *lac* repressor from *Escherichia coli*, LacI. Using a method that combines long-read and short-read DNA sequencing, we quantitatively measure the dose-response curves for nearly $10^5$ variants of the LacI genetic sensor. The resulting data provide a quantitative map of the effect of amino acid substitutions on LacI allostery and reveal systematic sequence-structure-function relationships. We find that in many cases, allosteric phenotypes can be quantitatively predicted with additive or neural-network models, but unpredictable changes also occur. For example, we were surprised to discover a new band-stop phenotype that challenges conventional models of allostery and that emerges from combinations of nearly silent amino acid substitutions.

**Keywords** allostery; genetic sensor; genotype-phenotype relationships; high-throughput measurements; transcription factor

**Subject Categories** Biotechnology & Synthetic Biology; Chromatin, Transcription & Genomics; Structural Biology

**Mol Syst Biol. (2021) 17: e10179**

## Introduction

Allostery is an inherent property of biomolecules that underlies cellular regulatory processes including sensing, signaling, and metabolism (Fenton, 2008; Motlagh *et al*, 2014; Razo-Mejia *et al*, 2018). With allosteric regulation, ligand binding at one site on a biomolecule changes the activity of another, often distal, site. Switching between active and inactive states provides a sense-and-response function that defines the allosteric phenotype. Quantitative descriptions relating that phenotype to its causal genotype would improve our understanding of cellular function and evolution, and advance protein design and engineering (Raman *et al*, 2014; He & Liu, 2016; Huang *et al*, 2016). However, the intramolecular interactions that mediate allosteric regulation are complex and distributed widely across the biomolecular structure, making the development of general quantitative descriptions challenging.

Recently described genotype-phenotype landscape approaches have enabled the phenotypic characterization of $10^4$–$10^5$ genotypes simultaneously (Li *et al*, 2016; Puchta *et al*, 2016; Sarkisyan *et al*, 2016; Domingo *et al*, 2018; Li & Zhang, 2018; Pressman *et al*, 2019). Measurements at this scale facilitate the exploration of genotypes with widely distributed mutations, making them ideal for probing complex biological mechanisms like allostery. However, to quantitatively characterize the sense-and-response phenotypes inherent to allostery, a measurement must encompass the full dose-response curve that describes biomolecular activity as a function of ligand concentration.

Genetic sensors have served as a model of allosteric regulation for decades, and today are central to engineering biology. Genetic sensors are allosteric proteins that regulate gene expression in response to stimuli, giving cells the ability to regulate their metabolism and respond to environmental changes. Like other allosteric biomolecules, the *lac* repressor, LacI, switches between an active state and an inactive state. In the active state, LacI binds to a DNA operator upstream of regulated genes, preventing transcription. Ligand binding to LacI stabilizes the inactive (non-operator-binding) state that allows transcription to proceed. This switching results in the allosteric phenotype that is quantitatively defined by a dose-response curve relating the concentration of input ligand ($L$) to the output response (the expression level of regulated genes, $G$). Genetic sensors typically have sigmoidal dose-response curves following the Hill equation:

$$G(L) = G_0 + \frac{G_\infty - G_0}{1 + \left(\frac{EC_{50}}{L}\right)^n}$$

where $G_0$ is basal gene expression in the absence of ligand, $G_\infty$ is gene expression at saturating ligand concentrations, $EC_{50}$ is the effective concentration of ligand that results in gene expression

---

1   National Institute of Standards and Technology, Gaithersburg, MD, USA
2   SLAC National Accelerator Laboratory, Menlo Park, CA, USA
3   Joint Initiative for Metrology in Biology, Stanford, CA, USA
    *Corresponding author. Tel: +1 301 975 2525; E-mail: david.ross@nist.gov

midway between $G_0$ and $G_\infty$, and the Hill coefficient, $n$, quantifies the steepness of the dose-response curve (Fig 1E).

As a framework to relate changes in the dose-response curve to the underlying biophysics of the LacI protein, we use recently

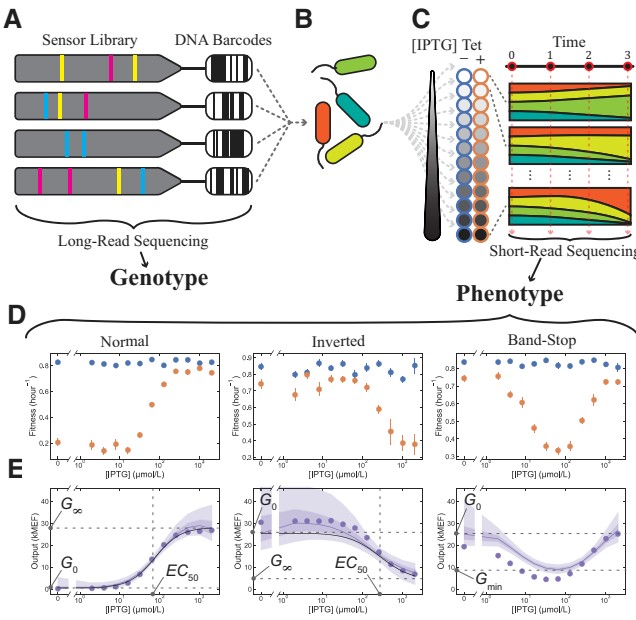

**Figure 1. Library-scale allosteric genotype-phenotype landscape measurement.**

A   A library of *lac* repressor (LacI) variants was generated by random mutagenesis of the *lacI* coding DNA sequence (CDS). The CDS for each variant was attached to a DNA barcode and inserted into a plasmid where the LacI variant regulated expression of a tetracycline resistance gene. The CDS and corresponding barcode on each plasmid were determined with long-read sequencing.

B   The library was transformed into *Escherichia coli*.

C   Cells containing the library were grown in 24 chemical environments, including 12 concentrations of the ligand IPTG, each with (orange) and without (blue) tetracycline. Cultures were maintained in exponential growth. Changes in the relative abundance of each variant were measured with short-read sequencing of DNA barcodes at four timepoints and were used to determine the fitness associated with each variant in each environment.

D   The fitness without tetracycline (blue) is independent of IPTG concentration. The fitness with tetracycline (orange) depends on the IPTG concentration via the dose-response of each variant. Error bars indicate ± one standard deviation estimated from least-squares fits of the barcode abundance vs time (Materials and Methods) and are often within markers. Data are from a single library-scale measurement.

E   Dose-response curves for 62,472 LacI variants were determined from the fitness measurements with Bayesian inference using a Hill equation model (black lines for variants with normal and inverted dose-response curves) and a Gaussian process (GP) model (purple lines, shaded regions indicate 50% and 90% credible intervals). Flow cytometry verification measurements (purple points) generally agreed with Bayesian inference results and verified the existence of the band-stop and other phenotypes. dose-response output was calibrated from fitness to fluorescent protein expression (Appendix Fig S10) and reported in molecules of equivalent fluorophore (MEF). Purple points represent the geometric mean of the YFP fluorescence minus the geometric mean of a zero-fluorescence control (92 MEF), as determined from a single flow cytometry measurement at each IPTG concentration.

described biophysical models that extend the general Monod-Wyman-Changeux (MWC) model of allostery (Monod *et al,* 1965) to the case of allosteric transcription factors (Daber *et al,* 2011; Razo-Mejia *et al,* 2018; Chure *et al,* 2019). Within those models, the dose-response curve depends on several biophysical parameters, including ligand-binding affinity, operator-binding affinity, and the allosteric constant, which is the equilibrium ratio between the inactive and active states in the absence of ligand and DNA operator (Monod *et al,* 1965; Daber *et al,* 2011; Razo-Mejia *et al,* 2018; Chure *et al,* 2019). The amino acid sequence (and corresponding structure) sets these biophysical parameters, and thus, amino acid substitutions can change these parameters (Daber *et al,* 2011; Chure *et al,* 2019). However, in the absence of data, the effect of any particular substitution on the biophysical parameters is unpredictable. Furthermore, substitutions distal to the active sites of a biomolecule can strongly affect allosteric function (Taylor *et al,* 2016; Leander *et al,* 2020). Consequently, to develop a more predictive understanding of allostery will require large-scale, quantitative measurements of changes to an allosteric dose-response curve resulting from widespread substitutions.

## Results

### Measuring the genotype-phenotype landscape

To measure the genotype-phenotype landscape for the allosteric LacI sensor, we first created a library of LacI variants using error-prone PCR and attached a DNA barcode to the coding DNA sequence (CDS) of each variant (Fig 1A). We used error-prone PCR across the full *lacI* CDS to investigate the effects of higher-order substitutions spread across the entire LacI sequence and structure. We then inserted the barcoded library into a plasmid where LacI regulates the expression of a tetracycline resistance gene (Appendix Fig S1A). Consequently, in the presence of tetracycline, the LacI dose-response modulates cellular fitness (i.e., growth rate) based on the concentration of the input ligand isopropyl-β-ᴅ-thiogalactoside (IPTG). We then transformed the library into *Escherichia coli* for the landscape measurement (Fig 1B). To ensure that most variants in the library could regulate gene expression, we used fluorescence-activated cell sorting (FACS) to enrich the library for variants with low $G_0$ (Appendix Fig S2). Then, using high-accuracy, long-read sequencing (Wenger *et al,* 2019), we determined the genotype for every variant in the library and indexed each variant to its attached DNA barcode (Fig 1A).

The library contained 62,472 different LacI genotypes, with an average of 7.0 single nucleotide polymorphisms (SNPs) per genotype. Many SNPs were synonymous, i.e., coded for the same amino acid, so the library encoded 60,398 different amino acid sequences with an average of 4.4 amino acid substitutions per variant (Appendix Fig S3B, the number of variants in the library at each mutational distance from the wild type are listed Appendix Table S1, and the number of observations of each amino acid substitution in the library is shown in Appendix Fig S4).

To quantitatively determine the allosteric phenotype for every LacI variant in the library, we developed a new method to characterize the dose-response curves for large genetic sensor libraries. Briefly, we grew *E. coli* containing the library in 24 chemical

environments (12 ligand concentrations, each with and without tetracycline). We used short-read sequencing of the DNA barcodes to measure the relative abundance of each variant at four timepoints during growth (Fig 1C). We then used the changes in relative abundance to determine the fitness associated with each variant in each environment (Fig 1D). Finally, for each variant in the library, we used the fitness difference (with vs without tetracycline) from all 12 ligand concentrations to quantitatively determine the dose-response curve using Bayesian inference (Fig 1E). Most variants had sigmoidal dose-response curves (e.g., Appendix Figs S5 and S6), which we analyzed using a Hill equation-based inference model to quantitatively determine the Hill equation parameters and their associated uncertainties. Some variants had non-sigmoidal dose-response curves (e.g., Appendix Figs S7 and S8, and discussion below), so we also analyzed all of the variants using a non-parametric Gaussian process (GP) inference model.

We compared the distributions of the resulting Hill equation parameters between two sets of variants: 39 variants with exactly the wild-type CDS for LacI (but with different DNA barcodes) and 310 variants with synonymous nucleotide changes (i.e., the wild-type amino acid sequence, but a non-wild-type DNA coding sequence). Using the Kolmogorov-Smirnov test, we found no significant differences between the two sets (P-values of 0.71, 0.40, 0.28, and 0.17 for $G_0$, $G_\infty$, $EC_{50}$, and $n$, respectively, Appendix Fig S9). So, for all subsequent analyses we considered only amino acid substitutions.

To evaluate the accuracy of the new method for library-scale dose-response curve measurements, we independently verified the results for over 100 LacI variants from the library. For each verification measurement, we chemically synthesized the CDS for a single variant and inserted it into a plasmid where LacI regulates the expression of a fluorescent protein (Appendix Fig S1B). We transformed the plasmid into *E. coli* and measured the resulting dose-response curve with flow cytometry (e.g., Fig 1E). We compared the Hill equation parameters from the library-scale measurement with those same parameters determined from flow cytometry measurements for each of the chemically synthesized LacI variants (Fig 2A–D). This served as a check of the new library-scale method's overall ability to measure dose-response curves with quantitative accuracy. The accuracy for each Hill equation parameter in the library-scale measurement was 4-fold for $G_0$, 1.5-fold for $G_\infty$, 1.8-fold for $EC_{50}$, and ± 0.28 for $n$. For $G_0$, $G_\infty$, and $EC_{50}$, we calculated the accuracy as: $\exp[\mathrm{RMSE}(\ln(x))]$, where $\mathrm{RMSE}(\ln(x))$ is the root-mean-square difference between the logarithm of each parameter from the library-scale and cytometry measurements. For $n$, we calculated the accuracy simply as the root-mean-square difference between the library-scale and cytometry results. The accuracy for the gene expression levels ($G_0$ and $G_\infty$) was better at higher gene expression levels (typical for $G_\infty$) than at low gene expression levels (typical for $G_0$), which is expected based on the non-linearity of the fitness impact of tetracycline (Appendix Figs S10 and S11). Measurements of the Hill coefficient, $n$, had high relative uncertainties for both barcode sequencing and flow cytometry, so the parameter $n$ was not used in any quantitative analysis. Overall, the flow cytometry results demonstrated that our experimental method measures dose-response curves with both high qualitative and quantitative accuracy (Fig 2A–D, Appendix Figs S5–S8).

## Effects of amino acid substitutions on LacI phenotype

During library construction, we chose the mutation rate to simultaneously achieve two objectives: exploration of a broad genotype-phenotype space, and acquisition of the single amino acid substitution data most useful for building quantitative biophysical models of allosteric function (Monod *et al*, 1965; Razo-Mejia *et al*, 2018; Chure *et al*, 2019). Starting from the wild-type DNA sequence for LacI, there were 2,110 possible SNP-accessible amino acid substitutions. Most of those substitutions were present in one or more variants within the library; however, nearly half were found only in combination with other substitutions. So, to comprehensively determine the impact of single amino acid substitutions, we constructed a deep neural network model (DNN) capable of accurately predicting the Hill equation parameters for LacI variants that were not directly measured. We tested two different neural network architectures: a recurrent DNN and a more conventional feed-forward DNN, as well as a linear-additive model. Of the three models, the recurrent DNN model provides the best predictive performance for each of the Hill equation parameters, though for $EC_{50}$, the recurrent DNN and linear-additive models have similar performance (Appendix Fig S12). So, for subsequent analysis, we used the recurrent DNN model, which captures the context dependence of amino acid substitution effects (Appendix Fig S12). In addition, to estimate uncertainties for the model predictions, we used approximate Bayesian inference methods as described in the Materials and Methods (Hochreiter & Schmidhuber, 1997).

We trained the DNN model to predict the Hill equation parameters $G_0$, $G_\infty$, and $EC_{50}$ (Appendix Fig S13), the three Hill equation parameters that were determined with relatively low uncertainty by the library-scale measurement. To evaluate the accuracy of the model predictions, we used the root-mean-square error (RMSE) for the model predictions compared with the measurement results. We calculated RMSE using only held-out data not used in the model training, and the split between held-out data and training data was chosen so that all variants with a specific amino acid sequence appear in only one of the two sets. For all three parameters, the RMSE for the model predictions increases with the number of amino acid substitutions relative to the wild type (Appendix Fig S14). Importantly, for single-substitution variants, the model RMSE is comparable to the experimental measurement uncertainty (Appendix Fig S15). So, we could confidently integrate the experimental and DNN results to provide a nearly complete map of the effects of SNP-accessible amino acid substitutions. Furthermore, by integrating information about the causal substitutions from multiple genetic backgrounds, the model provided improved estimates of $EC_{50}$ and $G_\infty$ for variants with $EC_{50}$ near or above the maximum ligand concentration measured (Appendix Fig S16).

The resulting map of single-substitution effects includes quantitative point estimates and uncertainties of the Hill equation parameters for 94% of the possible SNP-accessible amino acid substitutions (1,991 of 2,110; 964 directly from measured data, and 1,027 from DNN predictions; Appendix Figs S17–S19, Dataset EV1). Most of the 119 substitutions missing from the dataset were probably excluded by FACS during library preparation because they cause a substantial increase in $G_0$. These include 83 substitutions that have been shown to result in constitutively high $G(L)$ (Markiewicz *et al*, 1994; Pace *et al*, 1997). Of the 1,991 substitutions included in the dataset, 38%

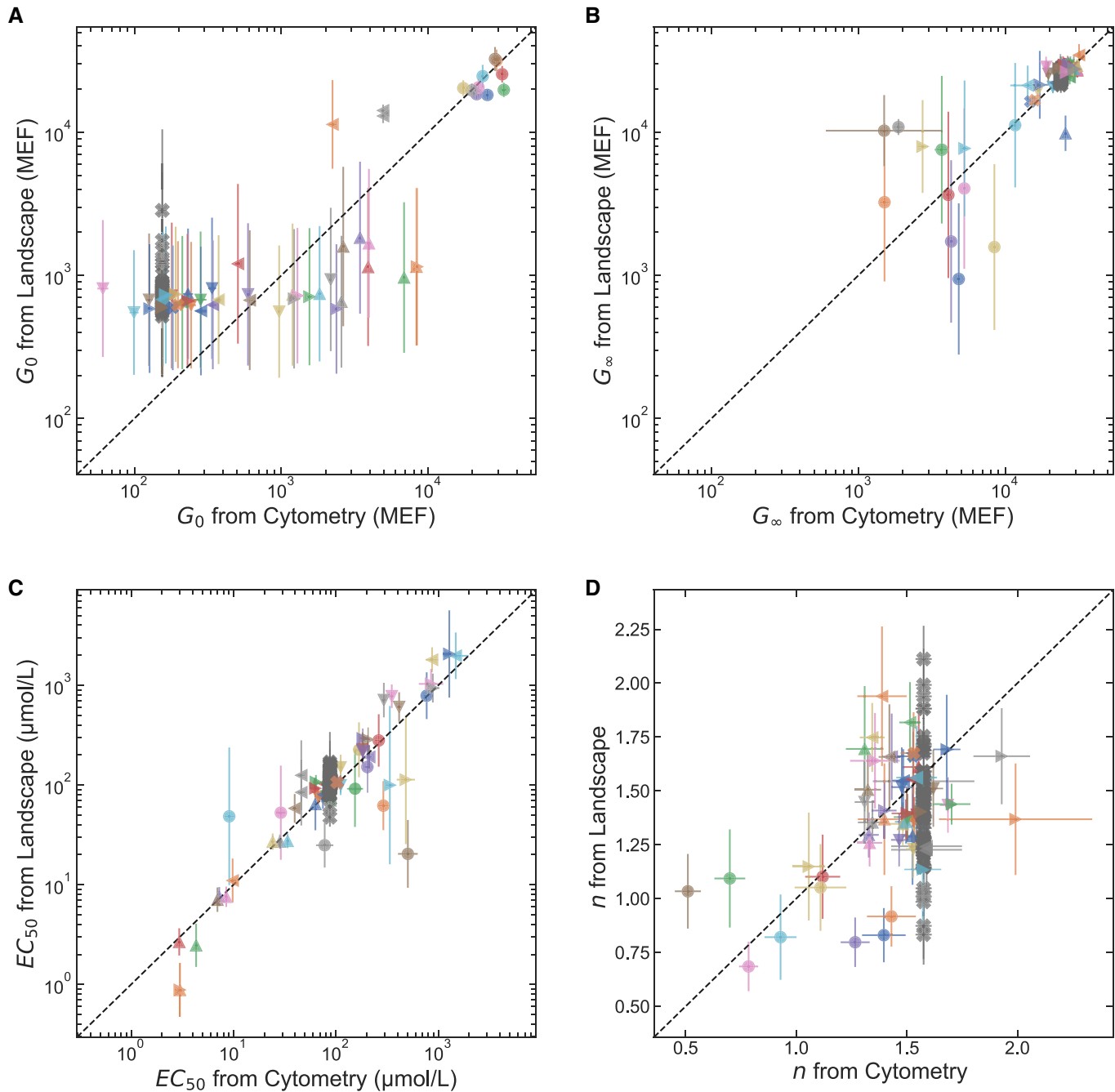

**Figure 2. Accuracy of the library-scale dose-response curve measurement.**

A–D The plots compare the results from the library-scale measurement (*y*-axis) with the flow cytometry verification results (*x*-axis) for each Hill equation parameter. Data are shown for all of the verified LacI variants with sigmoidal dose-response curves (i.e., band-stop and band-pass variants are not included). Data for different variants are plotted with different combinations of color and shape. Variants that occurred more than once in the library (with different DNA barcodes) are plotted multiple times. For example, the wild type (dark gray "X" symbols) is plotted 53 times. The accuracy for each Hill equation parameter is 4-fold for $G_0$ (A), 1.5-fold for $G_\infty$ (B), 1.8-fold for $EC_{50}$ (C), and $\pm$ 0.28 for $n$ (D). For $G_0$, $G_\infty$, and $EC_{50}$ (A–C), the accuracy is calculated as: $\exp(\mathrm{RMSE}(\ln(x)))$, where $\mathrm{RMSE}(\ln(x))$ is the root-mean-square difference between the logarithm of each parameter from the library-scale and cytometry measurements. For $n$, the accuracy is given simply as the root-mean-square difference between the library-scale and cytometry results. The inverse-variance-weighted coefficient of determination ($R^2$) for each Hill equation parameter is: 0.83 for $G_0$ (A), 0.55 for $G_\infty$ (B), 0.86 for $EC_{50}$ (C), and −0.04 for $n$ (D). The variance of the posterior distribution from the Bayesian inference was used for weighting. In addition, the contribution from the wild-type observations were weighted by a factor of 1/53 to avoid bias from multiple observations. In all plots, points indicate the median and error bars indicate $\pm$ one standard deviation from the Bayesian posterior. Data are from a single library-scale measurement, and a single flow cytometry measurement for each LacI variant at each IPTG concentration.

Source data are available online for this figure.

   

measurably affect the dose-response curve (beyond a 95% confidence bound).

The LacI protein has 360 amino acids arranged into three structural domains (Lewis *et al*, 1996; Flynn *et al*, 2003; Swint-Kruse *et al*, 2003). The first 62 N-terminal amino acids form the DNA-binding domain, comprising a helix–turn–helix DNA-binding motif and a hinge that connects the DNA-binding motif and the core domain. The core domain, comprising amino acid positions 63–324, is divided into two structural subdomains: the N-terminal core and the C-terminal core. The full core domain forms the ligand-binding pocket, core-pivot region, and dimer interface. The tetramerization domain comprises the final 30 amino acids and includes a flexible linker and an 18 amino acid α-helix (Fig 3, Appendix Table S2). Naturally, LacI functions as a dimer of dimers: Two LacI monomers form a symmetric dimer that further assembles into a tetramer (a dimer of dimers).

The effect of any amino acid substitution depends strongly on its location within the protein structure, indicating systematic sequence-structure-function relationships underlying LacI allostery (Fig 3). For example, substitutions that increase the basal expression, $G_0$, by more than 5-fold that were not excluded by FACS are located either in helix 4 of the DNA-binding domain, along the dimer interface, in the tetramerization helix, or at the protein start codon (Fig 3A and D). $G_0$ quantifies gene expression in the absence of ligand. So, within the biophysical models, substitutions that affect $G_0$ must alter either the operator-binding affinity, the allosteric constant, or the copy number of LacI proteins per cell (Daber *et al*, 2011; Razo-Mejia *et al*, 2018; Chure *et al*, 2019). Substitutions at the first and second codons (M1I, M1T, and, K2E) probably reduce the LacI copy number (Bivona *et al*, 2010; Hecht *et al*, 2017). But the other substitutions that affect $G_0$ (R51C, Q54K, L56M, T68N, S70C, L71Q, A92S, F226V, S322P, and Q352L) almost certainly change the operator-binding affinity, the allosteric constant, or both.

Interestingly, substitutions in helix 4 (R51C, Q54K, and L56M) that increase $G_0$ also decrease $EC_{50}$ approximately 10-fold, consistent with a change in the allosteric constant favoring the inactive state (Chure *et al*, 2019) (Appendix Fig S20A). Helix 4 forms part of the hinge connecting the DNA-binding motif to the core domain. It changes from a disordered coil to an order helix only upon binding of LacI to its cognate DNA operator, and interactions between the helix 4 residues of each LacI monomer have been shown to stabilize helix formation (Spronk *et al*, 1996) and therefore the active state of LacI. So, although helix 4 is more closely associated with the DNA-binding domain of LacI, the observed substitutions in helix 4 probably disrupt those interactions, changing the allosteric constant in a way that favors the inactive (non-operator-binding) state.

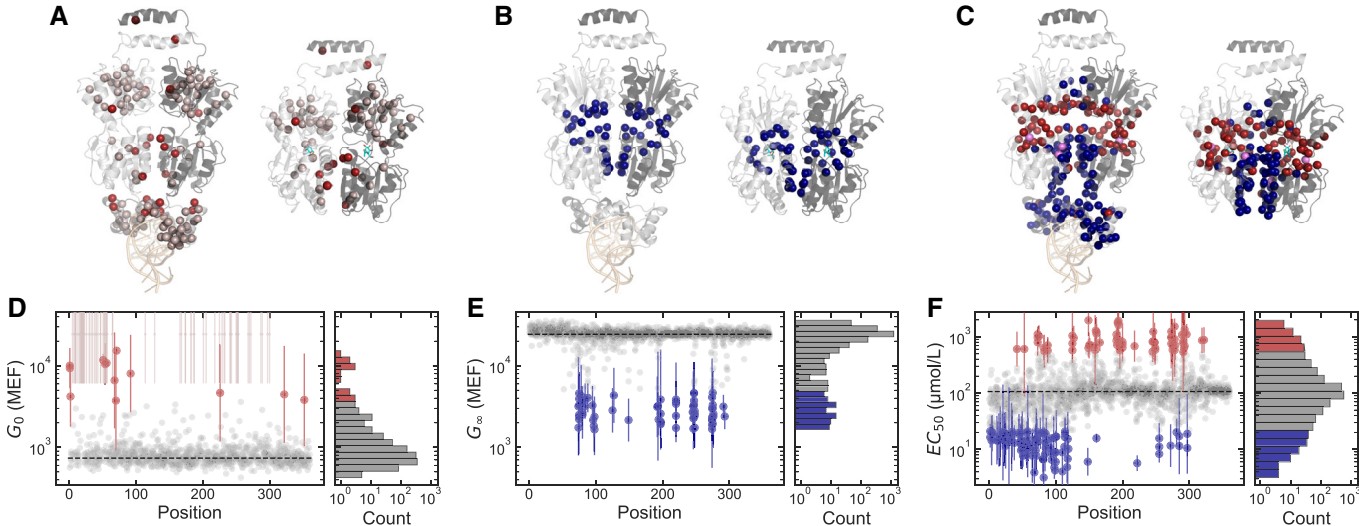

**Figure 3. Effect of single amino acid substitutions on allosteric function of LacI.**

A–C    Protein structures showing the locations of amino acid substitutions that affect each Hill equation parameter: $G_0$ (A), $G_\infty$ (B), $EC_{50}$ (C). For each, the operator-binding structure is shown on the left (operator DNA in light orange, PDB ID: 1LBG (Lewis *et al*, 1996)) and the ligand-binding structure is shown on the right (IPTG in cyan, PDB ID: 1LBH (Lewis *et al*, 1996)). Both structures are shown with the view oriented along the protein dimer interface, with one monomer in light gray and the other monomer in dark gray. Colored spheres highlight residues where substitutions cause a greater than 5-fold change in the Hill equation parameter relative to wild-type LacI. Red spheres indicate residues where substitutions increase the parameter, and blue spheres indicate residues where substitutions decrease the parameter. At three residues (A82, I83, and F161), some substitutions decrease $EC_{50}$, while other substitutions increase $EC_{50}$ (violet spheres in C).

D–F    Scatter plots showing the effect of each substitutions as a function of position. Substitutions that change the parameter by less than 5-fold are shown as gray points. Substitutions that change the parameter by more than 5-fold are shown as red or blue points with error bars. Histograms to the right of each scatter plot show the overall distribution of single-substitution effects.

Data information: In (A) and (D), gray-pink spheres and points indicate positions for substitutions that are completely missing from the library-scale dataset reported here and that have been shown by previous work to result in constitutively high $G(L)$ (Markiewicz *et al*, 1994; Pace *et al*, 1997). In (D–F), points show the best consensus estimate for the parameter values as described in the Materials and Methods. $G_0$ and $G_\infty$ are reported in molecules of equivalent fluorophore (MEF) based on the calibration with flow cytometry measurements (Materials and Methods). Error bars indicate ± one standard deviation estimated from the Bayesian posteriors. Data are from a single library-scale measurement.

Source data are available online for this figure.

The remaining substitutions that increase $G_0$ (T68N, S70C, L71Q, A92S, F226V, S322P, and Q352L) are far from the DNA-binding domain. So, they most likely affect the allosteric constant. Substitutions T68N, S70C, and L71Q, which are near the dimer interface, also decrease $EC_{50}$ between 4-fold and 30-fold (Appendix Fig S20A), similar to the substitutions in helix 4. Targeted molecular dynamic simulations have suggested that interactions between the L71 backbone and Q78′ (on the opposite monomer) stabilize the active state (Flynn et al, 2003). Substitutions at position L71 might disrupt these interactions, shifting the allosteric constant to favor the inactive state. The substitution L71Q, which replaces the hydrophobic leucine with a hydrophilic glutamine, causes the largest change (20-fold increase in $G_0$ and 14-fold decrease in $EC_{50}$), likely due to perturbation of the local hydrophobic environment at the dimer interface. Our results for hydrophobic substitutions at this position (L71V and L71M) support this picture, with just a 3-fold to 4-fold reduction in $EC_{50}$ (and little change to $G_0$), consistent with a smaller shift in the allosteric constant.

Approximately 3.5 and 5% of all amino acid substitutions decrease ligand-saturated expression, $G_\infty$, more than 5-fold or 2.5-fold, respectively. Substitutions that decrease $G_\infty$ by more than 5-fold are all located near the ligand-binding pocket or along the dimer interface (Fig 3B and E). Six of these substitutions also increase $EC_{50}$ more than 5-fold (A75T, D88N, S193L, Q248R, D275Y, and F293Y; Appendix Fig S20B). Except for D88N, which is at the dimer interface in helix 5, these substitutions are near the ligand-binding pocket. Substitutions near the ligand-binding pocket probably decrease ligand-binding affinity by changing the ligand-binding pocket environment directly. This would explain the observed increase in $EC_{50}$ for each of these substitutions, though studies with targeted substitutions have shown that substitutions near the ligand-binding pocket can also change the allosteric constant (Chure et al, 2019).

Amino acid substitutions that change the effective concentration, $EC_{50}$, are the most numerous and are spread throughout the protein structure, with approximately 9 and 20% of all substitutions causing a greater than 5-fold or 2.5-fold shift in $EC_{50}$, respectively (Fig 3C and F; Dataset EV1). The strongest effects are from substitutions in the DNA-binding domain, ligand-binding pocket, core-pivot region, or dimer interface.

Substitutions that cause the largest decrease in $EC_{50}$ are at the dimer interface and probably disrupt cross-dimer interactions. In particular, substitutions T68N (27-fold decrease) and L71Q (14-fold decrease) each probably disrupt the L71-Q78′ interaction (discussed above). Substitutions V99E (25-fold decrease), E100G (17-fold decrease), and V95M (16-fold decrease) are each in β-strand B and each probably disrupts the K84-K84′ interaction (discussed below). All of these substitutions likely shift the allosteric constant to favor the inactive state.

Substitutions that cause the largest increase in $EC_{50}$ are often near the ligand-binding pocket or core-pivot domain. Often, substitutions at these positions also affect $G_\infty$ (discussed above). However, we also identified nine positions near the ligand-binding pocket or core-pivot domain (N125, P127, D149, V192, A194, A245, N246, T276, Q291), where different substitutions either reduce $G_\infty$ by more than 5-fold or increase $EC_{50}$ by more than 5-fold, but not both (Dataset EV1). Given their positions, each of these substitutions probably disrupt the ligand-binding pocket thereby reducing

ligand-binding affinity, though they may also change the allosteric constant to favor the active state.

At three positions (A82, I83, and F161), different substitutions can either increase or decrease $EC_{50}$ more than 5-fold, depending on the substitution.

Residue F161 sits in the core-pivot region and is sequestered in a hydrophobic cluster (Swint-Kruse et al, 2001; Flynn et al, 2003), where the phenylalanine ring makes van der Waals contacts with Q291. In turn, Q291 is involved in hydrogen bonding networks that span the ligand-binding pocket and dimer interface (Flynn et al, 2003). During the transition between active and inactive states, the contacts between F161 and Q291 change, contributing to rearrangements throughout the LacI structure. At position F161, large hydrophobic amino acids (F161I, F161L) increase the $EC_{50}$ approximately 10-fold, while a slightly smaller hydrophobic amino acid (F161V) increases the $EC_{50}$ approximately 3-fold. In contrast, a small, hydrophilic amino acid (F161S) reduces $EC_{50}$ approximately 10-fold. The hydrophobic substitutions likely have little effect on the hydrophobic environment surrounding the position, but with different geometries, these amino acids may not make the required contacts with Q291. This could cause a shift in the allosteric constant to favor the active state, consistent with the observed increase in $EC_{50}$ for F161I, F161L, and F161V. On the other hand, the hydrophilic substitution at this position, F161S, likely disrupts the local hydrophobic environment, destabilizing the active state and shifting the allosteric constant to favor the inactive state, in agreement with the observed decrease in $EC_{50}$.

Positions A82 and I83 are in helix 5 of the N-terminal core domain, and both are proximal to and pointed toward helix 13. The A82E substitution, which replaces the diminutive alanine with the larger glutamate, decreases $EC_{50}$ approximately 30-fold. However, a smaller amino acid at this position (A82G) increases the $EC_{50}$ approximately 5-fold. These results suggest a steric clash between the side chain of residue 82 and helix 13 that is disrupts the active state and that effectively shifts the allosteric constant to favor the inactive state. At position I83, the I83F substitution decreases $EC_{50}$ approximately 5-fold while I83M increases $EC_{50}$ approximately 5-fold. Interestingly, both of these substitutions, as well as the wild-type isoleucine, are similar in volume (Zamyatnin, 1972) and hydropathy (Kyte & Doolittle, 1982). So, simple physiochemical differences do not satisfactorily account for the observed effects. The effects could perhaps be steric, as with position A82, but driven by changes in side-chain flexibility instead of size. Phenylalanine is the most rigid of the three side chains, followed by isoleucine, and the even more flexible methionine (Miao & Cao, 2016). As with position A82, our results suggest that such steric effects destabilize the active state, effectively shifting the allosteric constant to favor the inactive state.

We also identified five positions (H74, V80, K84, S97, M98) where different substitutions reduce either $G_\infty$ or $EC_{50}$ by more than 5-fold, but not both. These positions are all located at the dimer interface, specifically in or near helix 5 or β-strand B.

Substitutions at position H74 either decrease $EC_{50}$ approximately 8-fold (H74Q) or decrease $G_\infty$ approximately 10-fold while increasing $EC_{50}$ approximately 3-fold (H74P and H74Y). In the active state, residues H74 from both monomers form stable π-stacking interactions with each other. These interactions are disrupted in the inactive state, and instead, H74 forms a charge-charge interaction with

D278′ (on the opposite monomer) (Lewis *et al*, 1996). Substitutions at this position that abolish the π-stacking interactions would presumably destabilize the active state and shift the allosteric constant toward the inactive state. This is consistent with our result for H74Q. Our results for substitutions H74P and H74Y (decrease $G_\infty$ and increase $EC_{50}$) are consistent with either a shift in the allosteric constant to favor the active state or a decrease in the ligand affinity of the inactive state. H74Y can form the same π-stacking interactions seen in the active state of the wild type but cannot form the charge-charge interaction with D278′ to stabilize the inactive state. H74Y, therefore, would be expected to shift the allosteric constant toward the active state, agreeing with our observations. H74P cannot form either the π-stacking interactions or the charge-charge interaction with D278′, yet that substitution still increases $EC_{50}$ similarly to H74P. Since proline is a helix initiator (Richardson & Richardson, 1988; Kim & Kang, 1999) and H74 is positioned at the beginning of helix 5, H74P may shift the allosteric constant to favor the active state by stabilizing secondary structure.

Substitutions at positions K84, S97, and M98 decrease $G_\infty$ (S97P, M98R), or decrease $EC_{50}$ (K84N, S97W, M98L), or both (K84E, K84I, K84T, M98K). These residues are all involved in a coordinated process during the transition from the active state to the inactive state (Flynn *et al*, 2003). In the active state, the side chains of the K84 residues from both monomers sit in-plane with β-strand B and β-strand B′, interacting with the backbone of V94 and V96′ (both in β-strands B). In this process, K84 residues act as a bridge between the two β-strands. During the transition to the inactive state, K84 forms transient interactions with the side chain of S97 (also in β-strand B) and the backbone of M98, before eventually forming a stable charge-charge interaction with D88. Substitutions that disrupt this process have significant effects on the structure and function of LacI. For example, the substitution K84L causes significant structural changes to the N-terminal core domain and dimer interface (Bell *et al*, 2001), and substitutions at position S97 and M98 can greatly alter the biophysical properties of LacI (Zhan *et al*, 2010). Given the extent of structural and functional changes that can occur with substitutions involved in this process, precise mechanisms of the observed substitutions are difficult to predict, and observed changes are not easily described by the biophysical models. For example, within the biophysical models, to simultaneously decrease both $G_\infty$ and $EC_{50}$ (as observed for K84E, K84I, K84T, and M98K) requires a change to the ligand-binding affinity. Yet positions K84 and M98 are approximately 14 and 12 Å, respectively, from the ligand pocket (based on the wild-type LacI crystal structure).

None of the single amino substitutions measured in the library simultaneously decrease $G_\infty$ and increase $G_0$ (Appendix Fig S20C). This is not surprising, since substitutions that shift the biophysics to favor the active state tend to decrease $G_\infty$ while those that favor the inactive state tend to increase $G_0$, and the biophysical models (Daber *et al*, 2011; Razo-Mejia *et al*, 2018; Chure *et al*, 2019) indicate that only a combination of parameter changes can cause both modifications to the dose-response. The library did, however, contain several multi-substitution variants with simultaneously decreased $G_\infty$ and increased $G_0$. These inverted variants, and their associated substitutions are discussed below.

Combining multiple substitutions in a single protein almost always has a log-additive effect on $EC_{50}$. That is, the proportional

effects of two individual amino acid substitutions on the $EC_{50}$ can be multiplied together. For example, if substitution *A* results in a 3-fold change, and substitution *B* results in a 2-fold change, the double substitution, *AB*, behaving log-additively, results in a 6-fold change. Only 0.57% (12 of 2,101) of double amino acid substitutions in the measured data have $EC_{50}$ values that differ from the log-additive effects of the single substitutions by more than 2.5-fold (Fig 4). This result, combined with the wide distribution of residues that affect $EC_{50}$, reinforces the view that allostery is a distributed biophysical phenomenon controlled by a free energy balance with additive contributions from many residues and interactions, a mechanism proposed previously (Marzen *et al*, 2013; Motlagh *et al*, 2014) and supported by other recent studies (Leander *et al*, 2020), rather than a process driven by the propagation of local, contiguous structural rearrangements along a defined pathway.

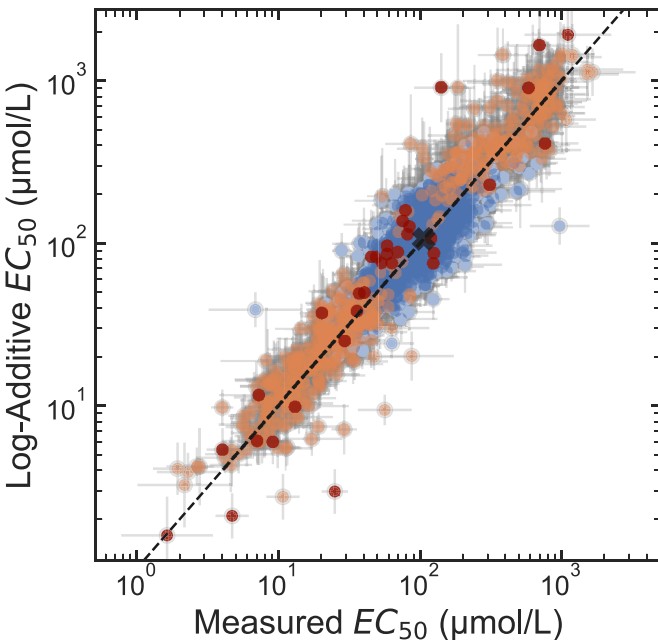

**Figure 4. The effects of amino acid substitutions on $EC_{50}$ of LacI are log-additive.**

The log-additive $EC_{50}$ for double-substitution LacI variants (i.e., two amino acid substitutions) was calculated assuming log-additivity of the effect of each single substitution on the $EC_{50}$ relative to wild-type LacI: log($EC_{50,AB}$/$EC_{50,wt}$) = log($EC_{50,A}$/$EC_{50,wt}$) + log($EC_{50,B}$/$EC_{50,wt}$), where "*wt*" indicates the wild type, "*A*" and "*B*" indicate the single-substitution variants, and "*AB*" indicates the double-substitution variant. The measured $EC_{50}$ of double-substitution variants is from the library-scale measurement. Orange points mark double-substitution variants in which one of the single substitutions causes a greater than 2.5-fold change in $EC_{50}$. Dark red points mark double-substitution variants in which both single substitutions cause a greater than 2.5-fold change in $EC_{50}$. The $EC_{50}$ of wild-type LacI is marked with a black "X". For this analysis, only experimental data were used (no results from the DNN model). Also, only data from LacI variants with low $EC_{50}$ uncertainty were used (SD(log$_{10}$($EC_{50}$)) < 0.35). Points show the best consensus estimate for the parameter values as described in the Materials and Methods. Error bars for the measured result indicate ± one standard deviation estimated from the Bayesian posteriors; error bars for the log-additive result indicate ± one standard deviation propagated from the Bayesian posterior uncertainties of the single-substitution results. Data are from a single library-scale measurement.

Source data are available online for this figure.

A similar analysis of log-additivity for $G_0$ and $G_\infty$ is complicated by the more limited range of measured values for those parameters, the smaller number of substitutions that cause large shifts in $G_0$ or $G_\infty$, and the higher relative measurement uncertainty at low $G(L)$. However, the effects of multiple substitutions on $G_0$ and $G_\infty$ are also consistent with log-additivity for almost every measured double-substitution variant (Appendix Fig S21).

Most of the non-silent substitutions discussed above are more likely to affect the allosteric constant than either the ligand or operator affinities. Within the biophysical models, those affinities are specific to either the active or inactive state of LacI, i.e., they are defined conditionally, assuming that the protein is in the appropriate state. So, almost by definition, substitutions that affect the ligand-binding or operator-binding affinities (as defined in the models) must be at positions that are close to the ligand-binding site or within the DNA-binding domain. Substitutions that modify the ability of the LacI protein to access either the active state or inactive state, by definition, affect the allosteric constant. This includes, for example, substitutions that disrupt dimer formation (dissociated monomers are in the inactive state), substitutions that lock the dimer rigidly into either the active or inactive state, or substitutions that more subtly affect the balance between the active and inactive states. Thus, because there are many more positions far from the ligand- and DNA-binding regions than close to those regions, there are many more opportunities for substitutions to affect the allosteric constant than the other biophysical parameters. Note that this analysis assumes that substitutions do not perturb the LacI structure too much, so that the active and inactive states remain somewhat similar to the wild-type states. Our results suggest that this is not always the case: consider, for example, the substitutions at positions K84 and M98 discussed above and the substitutions resulting in the inverted and band-stop phenotypes discussed below.

### Phenotypic innovation in an allosteric landscape

Beyond the comprehensive mapping of single-substitution effects, the LacI genotype-phenotype landscape measurement revealed a surprising number of variants with phenotypes that differ qualitatively from the wild type. For example, approximately 230 of the LacI variants have an inverted phenotype ($G_0 > G_\infty$, Fig 1E), accounting for approximately 0.35% of the measured library (Appendix Fig S3A). We verified the dose-response curves for 10 inverted variants with flow cytometry (e.g., Appendix Fig S6). To understand the mutational basis for the inverted phenotype, we examined a set of 43 strongly inverted variants (with $G_0/G_\infty > 2$, $G_0 > G_{\infty,wt}/2$, and $EC_{50}$ between 3 and 1,000 μmol/l). The results indicate that diverse substitutions can lead to the inverted phenotype. For example, we identified 10 amino acid substitutions associated with the inverted phenotype (S70I, K84N, D88Y, V96E, A135T, V192A, G200S, Q248H, Y273H, A343G, *P*-value < 0.005; Fig 5A and C; Appendix Table S3). However, none of these substitutions are present in more than 12% of the strongly inverted variants, and 51% of the strongly inverted variants have none of these substitutions. Furthermore, the set of strongly inverted variants are more genetically distant from each other than randomly selected variants from the library (Fig 5C, Appendix Fig S22). The genetic diversity of the inverted variants found in our measurement is striking when

compared with previous reports of inverted LacI variants resulting from site-saturated mutagenesis (Daber *et al*, 2011) or directed evolution with random mutagenesis (Poelwijk *et al*, 2011; Meyer *et al*, 2013). Those previous reports yielded only a small number of inverted variants with closely related genotypes and substitutions at specific positions that were key for inversion (I79, S97, and L296). Even more striking, most of the positions previously identified as important for the inverted phenotype are not significantly enriched in the set of strongly inverted sensors reported here.

The inverted LacI variants can provide specific insight into allosteric biophysics and structure–function relationships, since inversion of the dose-response curve requires inversion of both the allosteric constant and the relative ligand-binding affinity between the active and inactive states (Razo-Mejia *et al*, 2018; Chure *et al*, 2019). Although the set of strongly inverted LacI variants are genetically diverse, many of them have substitutions in similar regions of the protein that may account for the requisite biophysical changes (Appendix Table S3). First, 67% of the strongly inverted variants have substitutions within 7 Å of the ligand-binding pocket (compared with 31% of the full library, *P*-value = $1.15 \times 10^{-6}$), which likely contribute to the change in ligand-binding affinity. Surprisingly, 21% of the strongly inverted variants have no substitutions within 10 Å of the binding pocket, so binding affinity must be indirectly affected by distal substitutions in those variants. Second, nearly all strongly inverted variants have substitutions at the dimer interface (91%, compared with 54% for the full library, *P*-value = $2.05 \times 10^{-7}$), with most (70%) having substitutions in helix 5 (47%), helix 11 (28%), or both (5%, Fig 5A and C). This suggests that residues in those structural features are important for modulating the allosteric constant.

### Discovery of novel allosteric phenotypes

In addition to the inverted phenotypes, we were surprised to discover LacI variants with dose-response curves that did not match the sigmoidal form of the Hill equation. Specifically, we found variants with biphasic dose-response curves that repress or activate gene expression only over a narrow range of ligand concentrations. These include examples of LacI variants with band-stop dose-response curves (i.e., variants with high-low-high gene expression; e.g., Fig 1E, Appendix Fig S7), and LacI variants with band-pass dose-response curves (i.e., variants with low-high-low gene expression; e.g., Appendix Fig S8). Approximately 200 of the LacI variants have band-stop or band-pass phenotypes, accounting for approximately 0.3% of the measured library (Appendix Fig S3A). We verified the dose-response curves of 13 band-stop variants and two band-pass variants using flow cytometry (e.g., Appendix Figs S7 and S8). To our knowledge, this is the first identification of single-protein genetic sensors with band-stop dose-response curves.

Phenotypic similarities between band-stop and inverted LacI variants (i.e., high $G_0$, and initially decreasing gene expression as ligand concentration increases) suggest similar biophysical requirements (i.e., inversion of both the allosteric constant and the relative ligand-binding affinity between the two states). However, amino acid substitutions associated with the band-stop phenotype are remarkably different from those associated with inverted phenotype (V4A, A92V, G178D, H179Q, R195H, G265D, D292G, R351G, *P*-value < 0.005; Fig 5B and D; Appendix Table S4). While inverted

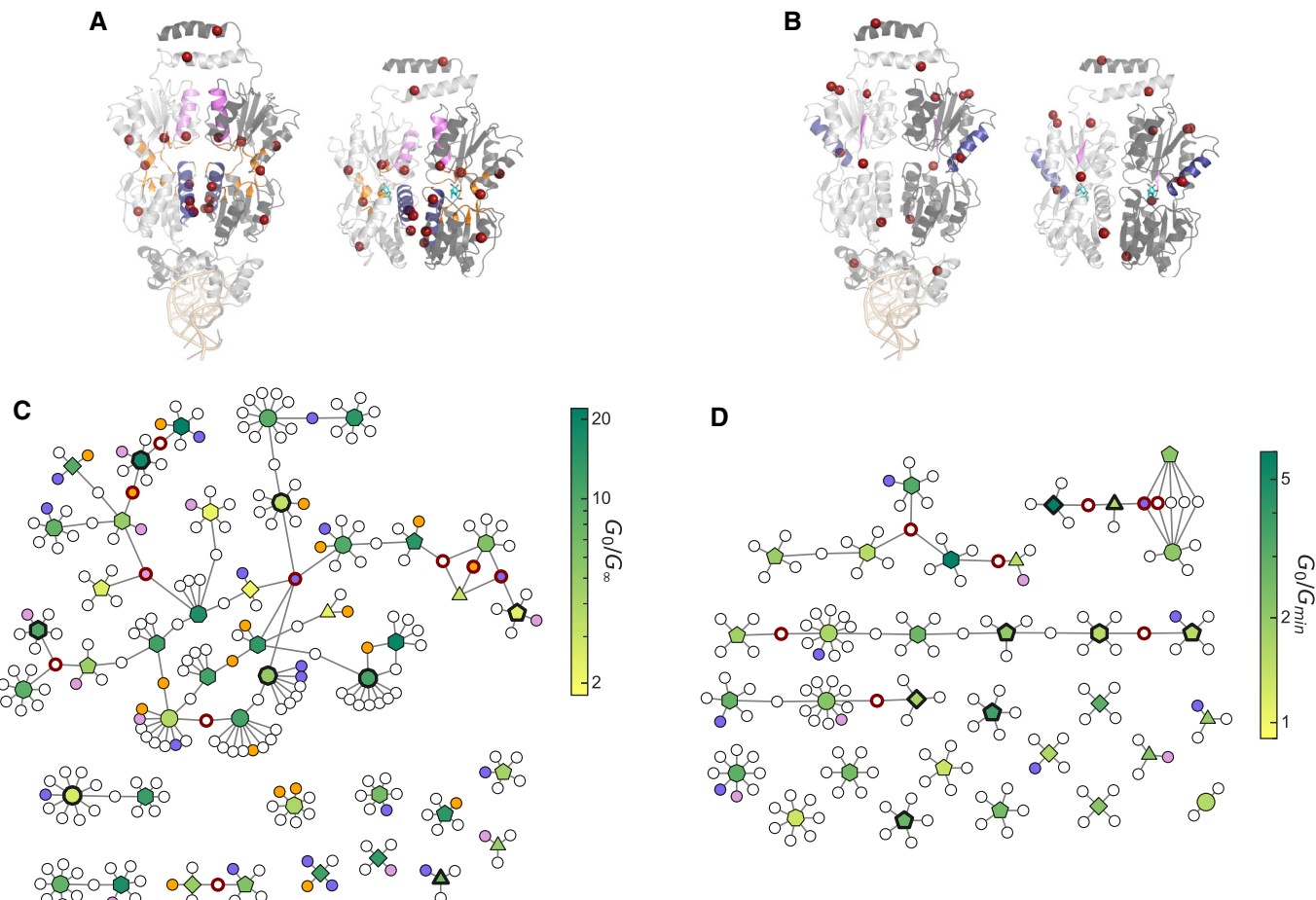

**Figure 5. Structural and genetic diversity of inverted and band-stop genotypes.**

A, B   Protein structures showing the locations of amino acid substitutions associated with strongly inverted (A) and strong band-stop (B) phenotypes. For each, the operator-binding structure of LacI is shown on the left (PDB ID: 1LBG), with the operator DNA at the bottom in light orange; the ligand-binding structure is shown on the right (PDB ID: 1LBH), with IPTG in cyan. Both structures are shown with the view oriented along the protein dimer interface, with one monomer in light gray and the other monomer in dark gray. The locations of associated (i.e., high-frequency) amino acid substitutions are highlighted as red spheres, and secondary structures where inverted or band-stop variants have amino acid substitutions at a significantly higher frequency than the full library are shaded with different colors. For strongly inverted variants (A), helix 5 is shaded blue, helix 11 is shaded violet, and the residues near the ligand-binding pocket are shaded orange. For strong band-stop variants (B), helix 9 is shaded blue, and β-strand J is shaded violet.

C, D   Network diagrams showing relatedness among genotypes for strongly inverted (C) and strong band-stop (D) variants. Within each network diagram, larger polygonal nodes represent LacI variants, with a colormap indicating the $G_0/G_\infty$ or $G_0/G_{min}$ ratio (see Fig 1E). The number of sides of the polygon indicates the number of amino acid substitutions relative to the wild type, and bold outlines indicate variants that were verified with flow cytometry. Smaller circular nodes represent specific amino acid substitutions, with connecting lines showing the substitutions for each variant. Bold red outlines on the substitution nodes indicate the associated substitutions shown as spheres in (A and B), and the shading of substitution nodes matches the shading used to highlight secondary structures in (A and B).

Source data are available online for this figure.

variants often have substitutions near the ligand-binding pocket and dimer interface, a set of 31 strong band-stop variants are twice as likely as the full library to have substitutions in helix 9 (32% compared with 16%, $P$-value = $2.14 \times 10^{-2}$) and nearly four times as likely to have substitutions in β-strand J (13% compared with 3.4%, $P$-value = $2.08 \times 10^{-2}$). Helix 9 is on the periphery of the protein, and β-strand J is in the center of the C-terminal core domain. Furthermore, 100% of the strong band-stop variants have substitutions in the C-terminal core of the protein, compared with 78% of the full library ($P$-value = $4.67 \times 10^{-4}$).

To further investigate the band-stop phenotype, we chose a strong band-stop LacI variant with only three amino acid substitutions (R195H/G265D/A337D). These three positions are distributed distally on the periphery of the C-terminal core domain, and the role that each of these substitutions plays in the emergence of the band-stop phenotype is unclear. To investigate the impact of these substitutions, we synthesized LacI variants with all possible combinations of those substitutions and measured their dose-response curves with flow cytometry. Although each single substitution resulted in a sigmoidal dose-response similar to wild-type LacI, the combination

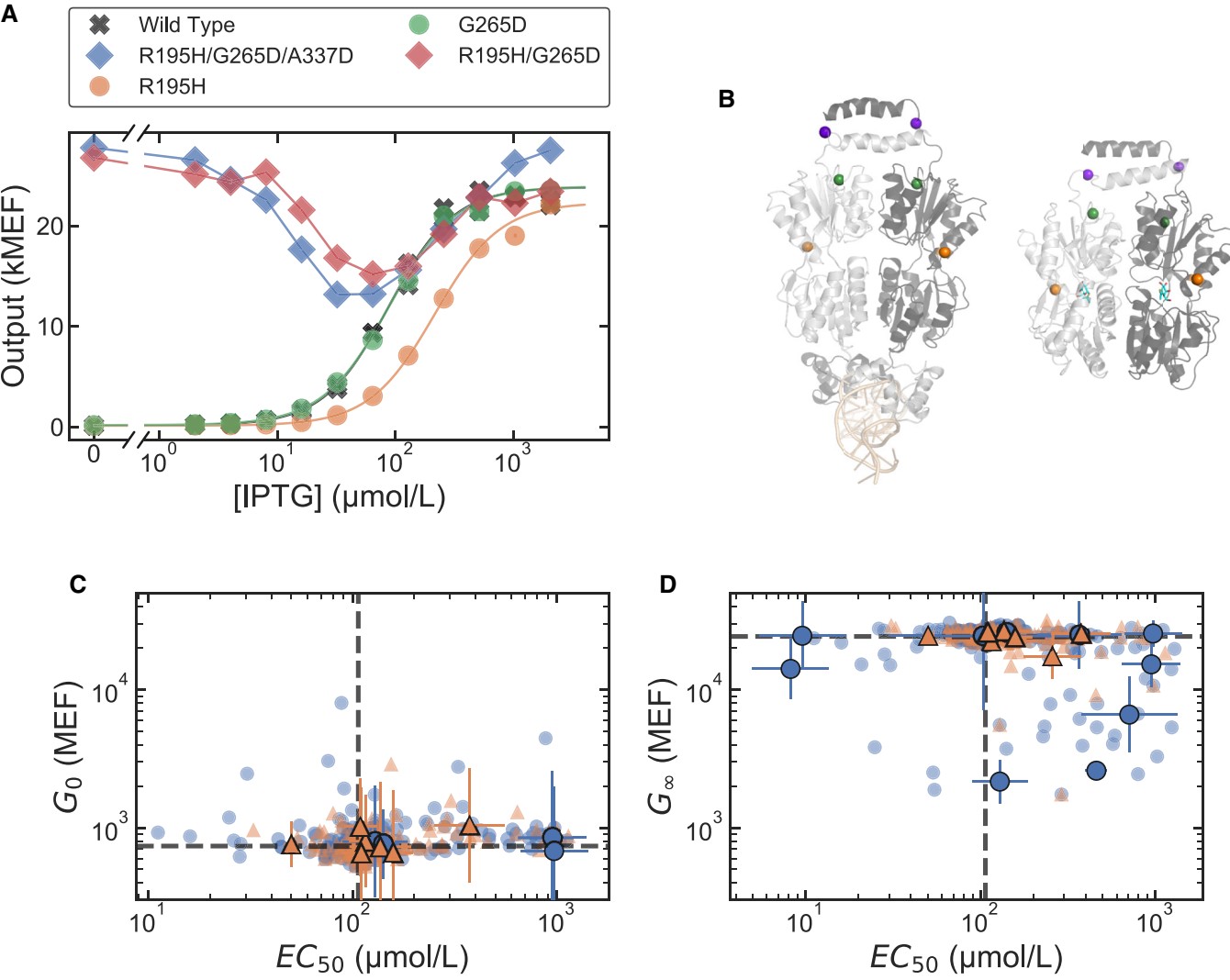

**Figure 6. The band-stop phenotype emerges from combinations of nearly silent amino acid substitutions.**

A   Dose-response curves measured with flow cytometry for selected LacI variants: wild-type LacI (gray "X"s), a strong band-stop variant identified from the library with only three amino acid substitutions (R195H/G265D/A337D; blue diamonds), LacI variants containing the single-substitution R195H (orange circles) and G265D (green circles), LacI variant with the double-substitution R195H/G265D (red diamonds). The single-substitution R195H (orange) or G265D (green) results in sigmoidal dose-response curves similar to wild-type LacI, but the combination of the two, R195H/G265D (red), results in a band-stop phenotype. The complete set of permutations of R195H, G265D, and A337D are shown in Appendix Fig S23.

B   Location of the three amino acid substitutions found in a strong band-stop variant. The operator-binding structure of LacI is shown on the left (PDB ID: 1LBG), with the operator DNA at the bottom in light orange; the ligand-binding structure is shown on the right (PDB ID: 1LBH), with IPTG in cyan. Amino acid positions R195 (orange), G265 (green), and A337 (purple) are highlighted as spheres.

C, D   Effects of individual amino acid substitutions associated with inverted and band-stop phenotypes. Each plot shows the joint effect of individual amino acid substitutions on two Hill equation parameters. The blue circles plotted with error bars show the effects of substitutions associated with the strongly inverted phenotype and the orange triangles plotted with error bars show the effects of substitutions associated with the strong band-stop phenotype. Most substitutions associated with the inverted phenotype cause a large shift in either $EC_{50}$, $G_\infty$, or both, consistent with the biophysical requirements for inverting the dose-response curve. In contrast, most of the amino acid substitutions associated with the band-stop phenotype are nearly silent. Light blue circles and light orange triangles show the effects for all amino acid substitutions found in the sets of strongly inverted and strong band-stop variants, respectively. Dashed gray lines mark the wild-type parameter values. Plotted data includes a combination of direct experimental measurements and DNN model predictions and is included in Dataset EV1. Error bars indicate ± one standard deviation estimated from the Bayesian posterior. Data are from a single library-scale measurement.

Source data are available online for this figure.

of two substitutions (R195H/G265D) gave rise to the band-stop phenotype (Fig 6A and B; Appendix Fig S23). To test whether this result applies to the band-stop phenotype generally, we used the single-substitution effects presented above to examine each of the substitutions associated with the strong band-stop phenotype. Individually, the substitutions associated with the band-stop phenotype

are nearly silent, i.e., they have little or no effect on the dose-response curve; yet in combination with other substitutions, they result in the band-stop phenotype. In contrast, most of the individual substitutions associated with the inverted phenotype cause a large shift in either $EC_{50}$, $G_\infty$, or both (Fig 6C and D).

## Discussion

For the goal of an improved understanding of allostery, our results reveal the dual nature of the problem: First, the DNN model and the mapping of single-substitution effects demonstrate that large-scale measurements and analysis can overcome the challenges inherent to the structural complexity of allosteric function. They can provide accurate predictions for specific allosteric proteins and can also reveal systematic sequence-structure-function relationships that may be more generalizable (i.e., the importance of the dimer interface and the log-additivity of $EC_{50}$). However, the band-stop phenotype highlights the limits of that predictability, as well as the constraints of conventional models of allostery.

While the allosteric function of many LacI variants is well described by extensions of the MWC model of allostery (Monod *et al*, 1965; Razo-Mejia *et al*, 2018; Chure *et al*, 2019), the band-stop phenotype is inconsistent with that model. In particular, the biphasic dose-response of the band-stop variants suggests negative cooperativity: that is, successive ligand-binding steps have reduced ligand-binding affinity. Negative cooperativity has been shown to be required for biphasic dose-response curves (Onufriev & Ullmann, 2004; Bouhaddou & Birtwistle, 2014). The biphasic dose-response and apparent negative cooperativity are also reminiscent of systems where protein disorder and dynamics have been shown to play an important role in allosteric function (Motlagh *et al*, 2014), including catabolite activator protein (CAP) (Popovych *et al*, 2006; Tzeng & Kalodimos, 2012) and the Doc/Phd toxin-antitoxin system (Garcia-Pino *et al*, 2010). This suggests that entropic changes may also be important for the band-stop phenotype. A potential mechanism is

that band-stop LacI variants have two distinct inactive states: an inactive monomeric state and an inactive dimeric state. In the absence of ligand, inactive monomers may dominate the population. Then, at intermediate ligand concentrations, ligand binding stabilizes dimerization of LacI into an active state which can bind to the DNA operator and repress transcription. When a second ligand binds to the dimer, it returns to an inactive dimeric state, similar to wild-type LacI. Similar dimerization-based regulation has been described before and supports the observed negative cooperativity and biphasic dose-response (Bouhaddou & Birtwistle, 2014). This mechanism and other possible mechanisms do not match the MWC model of allostery or its extensions (Monod *et al*, 1965; Daber *et al*, 2011; Razo-Mejia *et al*, 2018; Chure *et al*, 2019) and require a more comprehensive study and understanding of the ensemble of states in which these band-stop LacI variants exist.

Our most surprising and unpredictable result is the emergence of the band-stop phenotype from combinations of nearly silent amino acid substitutions. However, with over one hundred genetically diverse band-stop variants, our dataset provides a basis for more systematic understanding even in this case. Furthermore, the relatively high abundance of inverted and band-stop variants (approximately 0.35 and 0.2% of the library, respectively, Appendix Fig S3A) with genotypes near the wild type suggests that allosteric genotype-phenotype landscapes allow for rapid evolutionary innovation, a conclusion that is supported by the existence of natural transcription factors related to LacI with inverted phenotypes (Myers & Sadler, 1971; Rolfes & Zalkin, 1990).

Overall, our findings suggest that a surprising diversity of useful and potentially novel allosteric phenotypes exist with genotypes that are readily discoverable via large-scale landscape measurements. Novel phenotypes emerged at mutational distances greater than one amino acid substitution, highlighting the value in sampling a broader genotype space with higher-order mutations. Furthermore, the untargeted, random mutagenesis approach used here was critical for finding these novel phenotypes, as the genotypes required for these novel phenotypes were unpredictable.

## Materials and Methods

### Reagents and Tools table

| Reagent/Resource | Reference or source | Identifier or catalog number |
|---|---|---|
| **Experimental models** | | |
| *Escherichia coli* strain MG1655Δ*lac* | Sarkar *et al* (2020) | Addgene cat. #164844 |
| One Shot TOP10 Electrocomp *E. coli* | Invitrogen | Cat. #C404050 |
| **Recombinant DNA** | | |
| Plasmid pTY1 | This study | Addgene cat. #164831 |
| Plasmid pVER | This study | Addgene cat. #164830 |
| plasmid pUC19 | New England Biolabs | Cat. #N3041 |
| **Oligonucleotides and sequence-based reagents** | | |
| PCR primers for library construction | This study | Appendix Table S7 |
| Barcode PCR amplification primers | This study | Appendix Tables S8 and S9 |
| Paired-end adapter PCR amplification primers | This study | Appendix Table S10 |

**Reagents and Tools table**  (continued)

| Reagent/Resource | Reference or source | Identifier or catalog number |
|---|---|---|
| **Chemicals, enzymes, and other reagents** | | |
| Sera-mag SpeedBeads | GE Healthcare | Cat. #65152105050250 |
| GeneMorph II Random Mutagenesis Kit | Agilent | Cat. #200550 |
| Phusion Flash High-Fidelity PCR Master Mix | Thermo Scientific | Cat. #548L |
| T4 DNA Ligase | Thermo Scientific | Cat. #EL0011 |
| Calf Intestinal Phosphatase | New England Biolabs | Cat. #M0525L |
| FastDigest Buffer | Thermo Scientific | Cat. #B64 |
| FastDigest SgsI | Thermo Scientific | Cat. #FD1894 |
| FastDigest NheI | Thermo Scientific | Cat. #FD0974 |
| FastDigest XhoI | Thermo Scientific | Cat. #FD0695 |
| FastDigest ApaI | Thermo Scientific | Cat. #FD1414 |
| FastDigest DpnI | Thermo Scientific | Cat. #FD1703 |
| BspOI | Thermo Scientific | Cat. #ER2041 |
| XmaI | New England Biolabs | Cat. #R0180S |
| FseI | New England Biolabs | Cat. #R0588S |
| Gibson Assembly Master Mix | New England Biolabs | Cat. #E2611S |
| Rainbow Calibration Particles, 8 peaks | Spherotech | Cat. #RCP-30-20A |
| LB | BD Biosciences | Cat. #244620 |
| Casamino Acids | Fisher Bioreagents | Cat. #BP1424 |
| Bacto Tryptone | Gibco | Cat. #211705 |
| Bacto Yeast Extract | Thermo Scientific | Cat. #212750 |
| Glucose | Gibco | Cat. #2494001 |
| Glycerol | Thermo Scientific | Cat. #15514029 |
| Chloramphenicol | Fisher Bioreagents | Cat. #BP904-100 |
| Kanamycin | Thermo Scientific | Cat. #J1792406 |
| IPTG | Thermo Scientific | Cat. #R0393 |
| Tetracycline | Alfa Aesar | Cat. #B21408 |
| Neutralization Buffer | Qiagen | Cat. #19064 |
| Binding Buffer | Qiagen | Cat. #19066 |
| Rnase A | Qiagen | Cat. #19101 |
| Nuclease-free water | Thermo Scientific | Cat. #AM9938 |
| M9 Salts | BD Biosciences | Cat. #248510 |
| $CaCl_2$ | Fisher Bioreagents | Cat. #BP210-100 |
| $MgSO_4$ | Fisher Bioreagents | Cat. #BP213-1 |
| NaCl | Thermo Scientific | Cat. #AM9759 |
| KCl | Thermo Scientific | Cat. #AM9640G |
| $MgCl_2$ | Thermo Scientific | Cat. #AM9530G |
| NaOH | Millipore Sigma | Cat. #106462 |
| Tris–Cl, pH 7.5 | Fisher Bioreagents | Cat. #BP1757-100 |
| Tris–Cl, pH 8.0 | Invitrogen | Cat. #15-568-025 |
| EDTA | Fisher Bioreagents | Cat. #1311-200 |
| SDS | Millipore Sigma | Cat. #24802350 |
| Absolute Ethanol | Fisher Bioreagents | Cat. #BP2818500 |
| Tris–Cl, pH 8.5 | VWR | Cat. #MB-027-1000 |
| PEG-8000 | Sigma Aldrich | Cat. #89510 |

**Reagents and Tools table** (continued)

| Reagent/Resource | Reference or source | Identifier or catalog number |
|---|---|---|
| Focusing fluid | Invitrogen | Cat. #4488621 |
| PBS | Invitrogen | Cat. #AM9625 |
| Tween-20 | Fisher Bioreagents | BP337-100 |
| 1× TE Buffer | Thermo Scientific | Cat. #12090015 |
| 96-deep well plate | Eppendorf | Cat. #951033405 |
| 96-well plate | Abgene | Cat. #AB-1127 |
| 96-deep well plate | Eppendorf | Cat. #951033588 |
| 96-well DNA elution plate | Eppendorf | Cat. #30603303 |
| 96-well DNA binding plate | Nunc | Cat. #278010 |
| 96-well midi plate | Abgene | Cat. #AB-0765 |
| 96-well growth plate | 4titude | Cat. #4ti-0255 |
| Gas-permeable membrane | 4titude | Cat. #4ti-0598 |
| **Software** | | |
| Data analysis software | This study, Zhao et al (2018), and Schlecht et al (2017) | github.com/djross22/nist_lacI_landscape_analysis |
| **Other** | | |
| Qubit 1× dsDNA HS Assay Kit | Thermo Fisher Scientific | Cat. # Q33231 |
| QIAquick PCR Purification Kit | Qiagen | Cat. #28106 |
| QIAquick Gel Extraction Kit | Qiagen | Cat. #28115 |
| QIAprep Spin Miniprep Kit | Qiagen | Cat. #27106 |
| Laboratory automation system | Peak Analysis and Automation, S-Cell | Integrated components marked with (**) |
| Star liquid handler** | Hamilton | |
| a4S plate sealer** | 4titude | |
| Xpeel plate desealer** | Brooks | |
| Neo2SM plate reader** | BioTek | |
| Rotanta 460 Robotic Centrifuge** | Hettich Zentrifugen | |
| NGS Star liquid handler | Hamilton | |
| Illumina HiSeqX 300-cycle paired-end service | Novogene | |
| PacBio Sequel II | University of Maryland Institute of Genome Sciences | |
| Qubit 3 | Thermo Fisher Scientific | |
| Attune NxT Flow Cytometer | Thermo Fisher Scientific | |
| SH800S Cell Sorter | Sony | |

## Methods and Protocols

### Strain, plasmid, and library construction

All reported measurements were completed using *E. coli* strain MG1655Δ*lac* (Sarkar *et al*, 2020). Briefly, strain MG1655Δ*lac* was constructed by replacing the lactose operon of *E. coli* strain MG1655 (ATCC #47076) with the bleomycin resistance gene from *Streptoalloteichus hindustanus* (*Shble*).

Two plasmids were used for this work: a library plasmid (pTY1, Appendix Fig S1A) used for the measurement of the genotype and phenotype of the entire LacI library, and a verification plasmid (pVER, Appendix Fig S1B) used to verify the function of over 100 LacI variants from the library chosen to test the accuracy of the library-scale dose-response curve measurement method. A version of this protocol is maintained at protocols.io https://doi.org/10.17504/protocols.io.bjjxkkpn (preprint: Tack *et al*, 2020).

Plasmid pTY1 contained the *lacI* CDS and the lactose operator (*lacO*) regulating the transcription of a tetracycline resistance gene, *tetA*, which, in the presence of tetracycline, confers a measurable change in fitness connected with the expression level of the regulated genes. Plasmid pTY1 also encoded Enhanced Yellow Fluorescent Protein (YFP), which was used during library construction to select a library in which most of the LacI variants could function as allosteric repressors (see below).

Plasmid pVER contained a similar system in which LacI and *lacO* regulate the transcription of only YFP. Plasmid pVER was used to measure dose-response curves of clonal LacI variants using flow cytometry. Each variant chosen from the library for verification was chemically synthesized (Twist Biosciences), inserted into pVER, and transformed into *E. coli* strain MG1655Δ*lac* for flow cytometry measurements to confirm the dose-response curve inferred from the library-scale measurement.

The LacI library was generated by error-prone PCR of the wild-type *lacI* CDS encoded on plasmid pTY1. The library was constructed by splitting the lacI CDS into an N-terminal half and a C-terminal half using an ApaI restriction enzyme cut site that was near the center of the *lacI* CDS covering codons for A186, G187, and P188. Error-prone PCR of each half introduced genetic diversity, and then each unique sequence was attached to a DNA barcode. Plasmid-encoded versions of both sub-libraries were established (an N-terminal library, pNTL, and a C-terminal library, pCTL). These two sub-libraries were then assembled to generate the full *lacI* CDSs and to combine the two halves of the DNA barcode. The library was inserted into pTY1 along with randomly synthesized DNA barcodes (Appendix Fig S1A).

The plasmid containing the N-terminal library, pNTL, contained the N-terminal half of the *lacI* CDS (lacI-N) and a randomized nucleotide sequence that forms half of the DNA barcode. The protocol for constructing the N-terminal library was:

- PCR amplify the N-terminal half of the *lacI* CDS (coding for amino acids M1-A186) from pTY1 using the GeneMorph II Random Mutagenesis Kit (Agilent, cat. #200550) with primers DT.01 and DT.02 (Appendix Table S7). Agarose gel purify PCR product.
- PCR amplify the T7 terminator (TT7) with primers DT.03 and DT.04 with Phusion Flash polymerase (used for PCR unless otherwise specified) and then re-amplified with primers DT.03 and DT.05. Agarose gel purify PCR product.
- Assemble the two amplicons using assembly PCR with primers DT.02 and DT.03.
- Digest the assembled amplicon with restriction enzymes FastDigest ApaI and XmaI. Agarose gel purify.
- PCR amplify the pMB1 origin of replication and ampicillin resistance gene from plasmid pUC19 (NEB, cat #N3041S) with primers DT.06 and DT.07. Digest amplicon with restriction enzymes ApaI and XmaI and dephosphorylate the ends with Calf Intestinal Phosphatase. Agarose gel purify.
  o During this step, primer DT.06 incorporates half of the final DNA barcode, consisting of 27 randomized nucleotides interspersed with constant A/T bases to limit restriction site formation. Primer DT.06 was ordered with hand-mixed bases, with "N" representing equal ratios of A, T, G, and C.
  o Half DNA Barcode: 5′-TNNTNNNANNTNNNANNTNNNANNT NNNANNTNNNANNA-3′
- Ligate the assembly PCR product and the pUC19 amplicon using T4 DNA ligase (Thermo Scientific, EL0011) to form the final sub-library plasmid pNTL, and desalt using QIAquick PCR Purification Kit (Qiagen, cat. #28106).
- Transform into One Shot TOP10 Electrocomp *E. coli* (Invitrogen, cat. #C404050).
- Recover with SOC media at 37°C while shaking for 1 h.

The plasmid containing the C-terminal library, pCTL, contained the C-terminal half of the *lacI* CDS (lacI-C) and a randomized nucleotide sequence that forms the second half of the DNA barcode. The protocol for constructing the C-terminal library is:

- PCR amplify araC terminator from pTY1 using primers DT.08 and DT.09. Agarose gel purify PCR product.
- PCR amplify the C-terminal half of *lacI* CDS (coding for amino acids L189-Q360) from pTY1 using the GeneMorph II Random Mutagenesis Kit with primers DT.10 and DT.11. Agarose gel purify PCR product.
- Assemble the two amplicons using assembly PCR with primers DT.08 and DT.11.
- Digest the assembled amplicon with restriction enzymes FastDigest ApaI and XmaI. Agarose gel purify PCR product.
- PCR amplify the pMB1 origin of replication (pMB1) and ampicillin resistance gene from plasmid pUC19 with primers DT.12 and DT.13. Digest amplicon with restriction enzymes ApaI and XmaI dephosphorylate the ends with Calf Intestinal Phosphatase. Agarose gel purify.
  o During this step, primer DT.12 incorporates half of the final DNA barcode, consisting of 27 randomized nucleotides interspersed with constant A/T bases to limit restriction site formation. Primer DT.12 was ordered with hand-mixed bases, with "N" representing equal ratios of A, T, G, and C.
  o Half DNA Barcode: 5′-GNNTNNNANNTNNNANNTNNNTNNT NNNANNTNNNANNA-3′
- Ligate the assembly PCR product and the pUC19 amplicon using T4 DNA ligase to form the final sub-library plasmid pCTL, and desalt using the ligation product using QIAquick PCR Purification Kit.
- Transform into OneShot TOP10 Electrocomp *E. coli*.
- Immediately recover with SOC media at 37°C while shaking for 1 h.

Plasmids pNTL, pCTL, and pTY1 are starting points to assemble full length *lacI* CDS and combine the two halves of the DNA barcodes. The protocol is:

- PCR amplify the lacI-C and lacI-N libraries from plasmids pCTL and pNTL, respectively, both PCRs use primers DT.14 and DT.15.
- Digest the lacI-C library amplicon with restriction enzymes ApaI and DpnI and treat with CIP.
- Digest the lacI-N library amplicon with restriction enzymes ApaI and DpnI.
- Agarose gel purify both digested amplicons, then ligate the two together using T4 DNA ligase to assemble the sensor library with full length *lacI* CDS. Agarose gel purify the ligation product.
- Digest with FseI (NEB, cat. #R0588S). Agarose gel purify digested product.
- Circularize the linear product by ligating with T4 DNA ligase, then relinearize by digestion with restriction enzymes SgsI and NheI (Thermo Scientific, cat. #FD1894 and #FD0974) and agarose gel purify.
  o This is the assembled *lacI* CDS library with attached DNA barcodes, ready for ligation into pTY1.
- Prepare plasmid backbone by digesting pTY1 plasmid DNA with restriction enzymes SgsI and NheI, then treat with CIP. Agarose gel purify.
- Insert the assembled *lacI* CDS library with attached DNA barcodes into the pTY1 backbone using T4 DNA ligase with a 3-fold molar excess of pTY1 backbone.
- Desalt the ligation product and electroporate into MG1655Δ*lac*.

To prepare electrocompetent *E. coli* MG1655Δ*lac*:

- Dilute overnight culture of *E. coli* MG1655Δ*lac* 1,000-fold into 500 ml of LB media.

- Incubated the culture at 37°C for 3.5 h in a 2-l baffled Erlenmeyer flask to a final optical density at 600 nm (OD600) of approximately 0.8.
- Chill the culture in ice slurry for 20 min.
- Centrifuge the culture at 3,500 *g* for 10 min in refrigerated centrifuge at 4°C.
- Decant supernatant media, and then resuspended the cell pellet in 500 ml of 10% glycerol.
- Centrifuge the solution at 3,500 *g* for 10 min.
- Decant the supernatant glycerol solution.
- Repeated the glycerol wash one additional time (two washes total)
- Resuspended the cell pellet with residual 10% glycerol.
- Transform the plasmid-encoded sensor library (see above) into the freshly prepared electrocompetent MG1655Δlac.
- Immediately recover with SOC media at 37°C while shaking for 1 h.
- Dilute the library in LB media supplemented with glucose (2 g/l) and kanamycin (50 μg/ml) to a final volume of 500 ml and incubate for 12 h at 37°C while shaking.
- Divide the library into 1 ml aliquots and store them in 20% glycerol at −80°C (1:1 dilution with 40% glycerol).

Most of the variants in the initial library had high $G(0)$, i.e., the $\Gamma$ phenotype (Markiewicz *et al*, 1994). The initial library had a bimodal distribution of $G_0$, as indicated by flow cytometry results, with a mode at low fluorescence (near $G_0$ of wild-type LacI) and a mode at higher gene expression. To generate a library in which most of the LacI variants could function as allosteric repressors, we used fluorescence-activated cell sorting (FACS) to select the portion of the library with low fluorescence in the absence of ligand, gating at the trough between the two modes (Sony SH800S Cell Sorter, Appendix Fig S2). To allow comprehensive long-read sequencing of the library (PacBio sequel II, see Long-read sequencing section, below), we further reduced the library size by dilution of the FACS-selected library to create a population bottleneck of the desired size. For the work reported here, we used a library of approximately $10^5$ LacI variants (determined by serial plating and colony counting).

A spike-in control strain was used to normalize the DNA barcode read counts for the sequencing-based fitness measurement (see Library-scale fitness measurement section, below). The spike-in control strain contained the library plasmid (pTY1) with a LacI variant that had a constant, high *tetA* expression level. The fitness of the spike-in control was determined from $OD_{600}$ data acquired during growth of clonal cultures with the same automated growth protocol as used for the genotype-phenotype landscape measurement (see Growth protocol for landscape measurement section, below). The fitness of the spike-in control was measured in all 24 chemical environments and was independent of IPTG concentration but was slightly lower with tetracycline (0.75 h$^{-1}$) than without tetracycline (0.81 h$^{-1}$).

### Culture conditions

Unless otherwise noted, *E. coli* cultures were grown in a rich M9 media (3 g/l KH$_2$PO$_4$, 6.78 g/l Na$_2$HPO$_4$, 0.5 g/l NaCl, 1 g/l NH$_4$Cl, 0.1 mmol/l CaCl$_2$, 2 mmol/l MgSO$_4$, 4% glycerol, and 20 g/l casamino acids) supplemented with 50 μg/ml kanamycin.

*Escherichia coli* cultures were grown in a laboratory automation system that controlled preparation of 96-well culture plates with media and additives (i.e., IPTG and tetracycline). Cultures were grown in clear-bottom 96-well plates with 1.1 ml square wells (4titude, cat. #4ti-0255). The culture volume per well was 0.5 ml. Before incubation, an automated plate sealer (4titude, a4S) was used to seal each 96-well plate with a gas-permeable membrane (4titude, cat. #4ti-0598). Cultures were incubated in a multi-mode plate reader (BioTek, Neo2SM) at 37°C with a 1°C gradient applied from the bottom to the top of the incubation chamber to minimize condensation on the inside of the membrane. During incubation, the plate reader was set for double-orbital shaking at 807 cycles per minute. Optical density at 600 nm (OD600) was measured every 5 min during incubation, with continuous shaking applied between measurements. After incubation, an automated desealer (Brooks, XPeel) was used to remove the gas-permeable membrane from each 96-well plate.

### Growth protocol for landscape measurement

To measure the fitness and dose-response curve of every LacI variant in the library, a culture of *E. coli* containing the LacI library was mixed at a 99:1 ratio with a culture of the *E. coli* spike-in control. The culture was loaded into the automated microbial growth and measurement system (S-Cell, Peak Analysis and Automation) where it was distributed across a 96-well plate and then grown to stationary phase (12 h, Appendix Fig S24). Cultures were then diluted 50-fold into a new 96-well plate, Growth Plate 1, containing 11 rows with a 2-fold serial dilution gradient of IPTG with concentrations ranging from 2 to 2,048 μmol/l and one column without IPTG. Growth in IPTG allowed each variant to reach a steady-state tetA expression level in each IPTG concentration. Growth Plate 1 was grown for 160 min, corresponding to approximately 3.3 generations, and then diluted 10-fold into Growth Plate 2. Growth Plate 2 contained the same IPTG gradient as Growth Plate 1 with the addition of tetracycline (20 μg/ml) to alternating rows in the plate, resulting in 24 chemical environments, with each environment spread across 4 wells. Growth Plate 2 was grown for 160 min and then diluted 10-fold into Growth Plate 3, which contained the same 24 chemical environments as Growth Plate 2. This process was repeated for Growth Plate 4, which also contained the same 24 chemical environments. Each growth plate was pre-heated to 37°C before transferring the cells from the previous growth plate to avoid any disruption of cell growth due to large variations in temperature. The total growth time for the fitness measurements in the 24 chemical environments, 480 min across Growth Plates 2–4, corresponded to approximately 10 generations for the fastest-growing cultures. The 50-fold dilution factor from stationary phase into Growth Plate 1 and the 160-min growth time per plate were chosen to maintain the cultures in exponential growth for the entire 480 min. During each 160-min incubation, the cultures without tetracycline increased approximately 10-fold in optical density, to a final OD$_{600}$ of approximately 0.5 (corresponding to an estimated cell density of $4 \times 10^8$ cells/ml. Appendix Figs S25 and S26, Appendix Table S5). The protocol was:

- Inoculate 100 ml media in a 250-ml baffled Erlenmeyer flask with 2 ml frozen glycerol stock of library.
- Inoculate 50 ml media in a 250-ml baffled Erlenmeyer flask with scrapping of spike-in control glycerol stock.
- Incubate both at 37°C shaking at 300 rpm for 18 h.
- Into a 250-ml baffled Erlenmeyer flask, combine 49 ml of library

                                    

culture, 0.5 ml of spike-in control culture, and 50 ml of media, incubate 6 h at 37°C shaking at 300 rpm.

This mixture was used to begin the automated growth and measurement process:

- Distribute 450 µl media to each well of a 96-well growth plate (4titude, cat. #4ti-0255).
- Distribute 50 µl of culture mixture of library and spike-in control into each well of plate.
- Seal plate with a gas-permeable membrane and incubate in plate reader at 37°C for 12 h (BioTek, Neo2SM).
  o During incubations, the plate reader was set for continuous double-orbital shaking at 807 cycles per minute. $OD_{600}$ was measured every 5 min.
- Prepare Growth Plate 1:
  o Distribute 490 µl media across a 96-well growth plate
  o Use a 2-fold serial dilution of IPTG to add a gradient of IPTG across columns so that the final concentrations ranges from 2 to 2,048 µmol/l, and one column without IPTG.
- Ten minutes before the end of the 12-h incubation, preheat Growth Plate 1 to 37°C.
  o Preheat on temperature-controlled position set to 47°C for 10 min. Measurements of media temperature vs time indicated that this resulted in a media temperature of approximately 37°C.
- Remove 12 h growth plate from plate reader, remove gas-permeable membrane, and transfer 10 µl of culture from each well into the corresponding well of Growth Plate 1.
- Seal Growth Plate 1 with a gas-permeable membrane and incubate in plate reader at 37°C for 160 min.
  o Growth Plate 1 contains only the IPTG gradient (no tetracycline). This allows cells to reach exponential growth and to reach steady-state expression of tetA before adding tetracycline.
- Prepare Growth Plate 2
  o Distribute media across a 96-well growth plate, 450 µl total volume.
  o In alternating rows, supplement media with tetracycline to a final concentration of 20 µg/ml (rows B, D, F, H).
  o Use a 2-fold serial dilution of IPTG to add a gradient of IPTG across columns so that the final concentrations ranges from 2 to 2,048 µmol/l, and one column without IPTG.
- Ten minutes before the end of the 160-min incubation, preheat Growth Plate 2 to 37°C.
  o Preheat on temperature-controlled position set to 47°C for 10 min. Measurements of media temperature vs time indicated that this resulted in a media temperature of approximately 37°C.
- Remove Growth Plate 1 from plate reader, remove gas-permeable membrane, and transfer 50 µl of culture from each well of Growth Plate 1 into the corresponding well of Growth Plate 2.
- Seal Growth Plate 2 with a gas-permeable membrane and incubate in plate reader at 37°C for 160 min.
- Immediately proceed with plasmid DNA extraction for Growth Plate 1 (below).
- Repeat plate preparation and dilution protocol for Growth Plate 3 and Growth Plate 4 (with the same IPTG gradient and tetracycline in rows B, D, F, H).
- At the conclusion of Growth Plate 4 proceed with plasmid DNA extraction, there is no dilution into another plate.

After each growth plate was used to seed the subsequent plate (or at the end of 160 min for Growth Plate 4), the remaining culture volumes for each chemical environment (approximately 450 µl/well, four wells per plate) were combined and pelleted by centrifugation (3,878 $g$ for 10 min at 23°C). Plasmid DNA was then extracted from the 24 combined samples with a custom method using reagents from the QIAprep Miniprep Kit (Qiagen cat. #27104) on an automated liquid handler equipped with a positive-pressure filter press (a version of the protocol is maintained at protocols.io https://doi.org/10.17504/protocols.io.bjjvkkn6) (preprint: Alperovich et al, 2020a). The protocol was:

- Resuspend each cell pellet in in 200 µl of resuspension buffer (50 mmol/l of Tris–CL pH 8.0, 10 mmol/l EDTA, 100 µg/ml RNase A).
- Transfer resuspended cell samples to a new 96-well plate (Abgene, cat. #AB-1127) located on an automated microplate shaker.
- Add 250 µl lysis buffer (200 mmol/l NaOH, 10 g/l SDS) to each sample and mix by shaking the plate at 90 rpm for 2 min.
- Add 350 µl cold (4°C) Neutralization Buffer (Qiagen, cat. #19064) to each sample and mix by shaking at 90 rpm for 2 min.
- Using wide bore tips (3.2 mm tip diameter), gently mixed the samples by three repeated cycles of aspiration and dispensing, then transfer samples to a 96-well filter plate (Agilent, Cat. #201702-100) and allowed to settle in the filter plate for 2 min.
- Use the filter press to push the lysate solutions through the filter plate into a new 96-well deep well plate (Eppendorf, cat. #951033588) at 20 psi for 180 s followed by 65 psi for 30 s.
- Transfer the cleared lysate solutions to a 96-well glass fiber binding plate (Nunc, cat. #278010) and use the filter press to push the solutions through the binding plate at 40 psi for 60 s.
- Add 900 µl Binding Buffer (Qiagen, cat. #19066) to each well and use the filter press to push buffer through the binding plate at 40 psi for 60 s.
- Add 900 µl Wash Buffer (8 mmol/l Tris–Cl, pH7.5, 80% ethanol) to each well and use the filter press to push buffer through the binding plate at 40 psi for 60 s.
- Use the filter press to dry the binding plate by applying 65 psi for 7 min.
- Add 100 µl nuclease-free water (Thermo Scientific, cat. #AM9938) warmed to 60°C to each well; wait for 5 min.
- Elute DNA from binding plate into a 96-well low-binding elution plate (Eppendorf, cat. #30603303) using the filter press at 65 psi for 7 min.

Supernatant removal, cell resuspension, and cell sample transfer were performed using the automated liquid handler's 8-channel head with which each channel is capable of independent movement and liquid-level sensing to allow for variations in cell culture volume and density recovered from each sample. Most of the subsequent pipetting was performed using the automated liquid handler's 96-channel head with an offset pickup of 24 pipette tips so that the timing for each sample was identical. Pipetting the water for elution was performed using the automated liquid handler's 8-channel head to allow for individual channel movement.

For the DNA extracted from Growth Plate 4, the filter press jammed just before elution, so those samples were eluted by centrifugation at 1,000 $g$ for 3 min.

After elution, the concentration of DNA in each sample ranged from undetectable up to approximately 1.5 ng/μl. This corresponds to an estimated maximum of $10^{10}$ plasmids per sample.

### Barcode sequencing

After plasmid extraction, each set of 24 plasmid DNA samples was prepared for barcode sequencing. Briefly, the plasmid DNA was linearized with ApaI restriction enzyme. Then, a three-cycle PCR was performed to amplify the barcodes from the plasmids and attach sample multiplexing tags so samples from different chemical environments could be distinguished when pooled and run on the same sequencing flow cell. Eight forward index primers (Appendix Table S8) and 12 reverse index primers (Appendix Table S9) were used to label the amplicons from each sample across the 24 chemical environments and the four time points. After a magnetic bead-based cleanup step, a second, 15-cycle PCR was run to attach the standard Illumina paired-end adapter sequences and to amplify the resulting amplicons for sequencing (Appendix Table S10). After a second magnetic bead-based cleanup, the 24 samples from each time point were pooled and stored at 4°C until sequencing. This entire process was completed using a Hamilton NGS-STAR-automated liquid handler programmed with a custom sequencing sample preparation method with the following steps (a version of the protocol is maintained protocols.io https://doi.org/10.17504/protocols.io.bjjzkkp6) (preprint: Alperovich et al, 2020b). The protocol was:

- Prepare Sera-Mag SpeedBeads Carboxyl Magnetic Beads:
  - To a 50 ml conical tube, add 9 g PEG-8000, 10 ml of 5 mol/l NaCl, 500 μl of 1 mol/l Tris–HCl, pH 8, 500 μl of 0.5 mol/l EDTA.
  - Add water to a final volume of 45 ml and mix until PEG dissolves.
  - Add 27.5 μl Tween-20 and mix.
  - Vortex Sera-mag SpeedBeads to resuspend and transfer 1 ml to each of two 2 ml microcentrifuge tubes.
  - Place SpeedBeads solutions on a magnetic separation rack until beads are drawn to the magnet and solutions are clear.
  - Remove supernatant.
  - Add 1 ml 1× TE Buffer to each microcentrifuge tube containing beads, remove from magnetic stand, and vortex.
  - Repeat the TE wash step two additional times, then resuspend in 1 ml 1× TE buffer,
  - After mixing, add both 1 ml SpeedBeads solutions to the 50 ml conical tube containing the PEG solution.
  - Add water to a final volume of 50 ml and mix gently.
  - Store in the dark at 4°C until use.
- Transfer 35 μl of each plasmid DNA sample to a PCR plate (Bio-Rad, HSP9645).
- Add 4 μl FastDigest Buffer and 1 μl ApaI Restriction Enzyme Solution for plasmid DNA linearization; mix 3× by repeated aspiration and dispense.
  - Prepare a mixture of 4:1 of FastDigest Buffer:ApaI beforehand to expedite process.
- Incubate samples in PCR plate in automated thermocycler at 37°C for 15 min.
- Add 2.5 μl Forward Index Primer and 2.5 μl Reverse Index Primer to each sample (20 μmol/l stock primer solutions, primer sequences are listed in Appendix Tables S7 and S8).

- The Forward and Reverse Index Primers attach sample multiplexing tags to the resulting amplicons so that the different samples can be distinguished when they are pooled and run on the same Illumina platform sequencing flow cell. Eight different Forward Index Primers and 12 different Reverse Index Primers are used to uniquely label the amplicons from each sample across the 24 different chemical conditions and the four Growth Plates used for barcode sequencing.
- Add 45 μl Phusion Flash 2× Master Mix to each sample; mix 3× by repeated aspiration and dispense.
- Run the first PCR in automated thermocycler with the following conditions:
  - Initial denaturation: 98°C for 60 s.
  - Three cycles:
    - Denaturation: 98°C for 10 s.
    - Annealing: 58°C for 20 s.
    - Elongation: 72°C for 20 s.
  - Final extension: 72°C for 60 s.
  - Cooling: 20°C for 15 s.
- During the first PCR, pipette 48 μl Magnetic Bead/PEG-NaCl Stock into each of 24 wells in a 96-well midi plate (Abgene, cat. #AB-0765); mix bead stock thoroughly by repeated aspiration and dispense before each transfer.
- When the first PCR is finished, transfer 80 μl from each PCR reaction to a well in the midi plate; mix PCR solution and bead stock 10× by repeated aspiration and dispense; wait for 5 min.
  - During this step the ratio of Bead Stock to PCR volume is 0.6×. Consequently, because of the relatively low PEG concentration in the mixture, the 5,500 bp plasmid template binds to the beads.
- Move the midi plate to 96-well magnet base to pull the magnetic beads out of suspension; wait for 2 min.
- Transfer 128 μl of supernatant from each sample to a clean well in the midi plate (still on the magnet base); wait an additional 3 min for final bead separation.
- Transfer 118 μl of supernatant (containing the 218 bp PCR amplicon and primers) from each sample to another clean well in the midi plate; move the midi plate from the magnet base to automated shaker.
- Add 70.8 μl Magnetic Bead Stock (0.6× of 118 μl supernatant volume) to each sample; mix supernatant and bead stock 10× by repeated aspiration and dispense; wait for 5 min. During this step, only the 206 bp PCR amplicon binds to the beads; the 70 bp primers remain in the supernatant.
- Move the midi plate back to the magnet base; wait for 5 min.
- Remove and discard the supernatant from each well.
- Add 200 μl 80% ethanol to each magnetic bead pellet; wait 30 s.
- Remove and discard the ethanol supernatant; then, using 50 μl tips, remove residual supernatant from the bottom of each well.
- Move the midi plate from magnet base to the automated shaker; allow magnetic bead pellets to dry for 5 min at room temperature.
- Add 35 μl nuclease-free water to each sample; resuspend beads by 5× repeated aspiration and dispense.
- Mix samples by shaking at 1,800 rpm for 10 s; wait for 5 min.
- Move the midi plate back to the magnet base; wait for 5 min.
- During 5-min wait, pipette 33 μl Phusion Flash 2× Master Mix and 1.5 μl of each secondary PCR primer (20 μmol/l) into each of 24 wells in a PCR plate.

- After 5-min wait, transfer 30 μl of each supernatant from the midi plate (on magnet base) to the PCR plate.
- Run the second PCR in the automated thermocycler with the following conditions:
  - Initial denaturation: 98°C for 60 s.
  - 15 cycles:
    - Denaturation: 98°C for 10 s.
    - Annealing and elongation: 72°C for 30 s.
  - Final extension: 72°C for 120 s.
  - Cooling: 15°C for 20 s.
- During the second PCR, pipette 66 μl (1.1× of the 60 μl PCR volume) Magnetic Bead Stock into each of 24 clean wells in the midi plate (on the automated shaker); mix bead stock well by repeated aspiration and dispense before each transfer.
- When the second PCR is finished, transfer 60 μl from each PCR to a well in the midi plate; mix the PCR solution and bead stock 10× by repeated aspiration and dispensing; wait for 5 min.
  - During this step the ratio of Bead Stock to PCR volume is 1.1, the 306 bp PCR amplicon binds to the beads while the 70 bp primers remain in the supernatant.
- Move the midi plate back to the magnet base; wait for 4 min.
- Remove and discard the supernatant from each well.
- Add 200 μl 80% ethanol to each magnetic bead pellet; wait 30 s.
- Remove and discard the ethanol supernatant.
- Move the midi plate from the magnet base to the automated shaker; allow magnetic bead pellets to dry for 5 min at room temperature.
- Add 50 μl Elution Buffer (10 mmol/l Tris–Cl, pH 8.0) to each sample; resuspend beads by 5× repeated aspiration and dispense.
- Mix samples by shaking at 1,800 rpm for 10 s; wait for 5 min.
- Move the midi plate back to the magnet base; wait for 5 min.
- Transfer the supernatant from each of the 24 samples to a single pooled sequencing sample, one pooled sample per input growth plate.

After each 24-sample sequencing sample preparation, we transferred the resulting pooled samples to a 2-ml microcentrifuge tube and placed the tube on a magnetic separation rack until the remaining beads were drawn to the magnet (approximately 5 min). We then transferred the fully clarified samples to a new 2-ml microcentrifuge tube and stored them at 4°C. We stored the pooled sequencing samples for each growth plate separately until needed for sequencing. The concentration of DNA in each pooled sequencing sample ranged from 10 to 16 ng/μl. DNA concentrations throughout were determined by fluorimetry (Thermo Fisher Scientific, Cat. # Q33231).

For sequencing, DNA was diluted to 5 nmol/l and combined with 20% phiX control DNA. DNA from each of the 4 time points was sequenced in a separate lane on an Illumina HiSeqX using paired-end mode with 150 bp in each direction.

To count DNA barcodes and estimate the fitness associated with each LacI variant, the sequencing data were analyzed using custom software written in C# and Python, and the Bartender1.1 barcode clustering algorithm (Zhao *et al,* 2018) (https://github.com/djross22/nist_lacI_landscape_analysis).

The sequence of the nominal Illumina compatible amplicon was (with Illumina adapters and flow cell binding sequences in italics):

*AATGATACGGCGACCACCGAGATCTACACTCTTTCCCTACACGACGC*
*TCTTCCGATCT*ZZZZZZZZZXXXXXXXXXXX<u>CATC</u>GGTGAGCCCGGGCT
GTC**GGCGT**NNTNNNANNTNNNANNTNNNANNTNNNANNTNNN
ANN**ATATG**CCAGCAGGCCGGCC**ACGCT**NNTNNNANNTNNNANN
ANNNANNTNNNANNTNNNANN**CGGTG**GCCCGGGCGGCCGCAC
GATGCGTCCGGCGTA<u>GAGG</u>XXXXXXXXXXXZZZZZZZZZ*AGATCGGAA*
*GAGCGGTTCAGCAGGAATGCCGAGACCGATCTCGTATGCCGTCTTCT*
*GCTTG*

The nominal forward and reverse reads from paired-end barcode sequencing were:

ZZZZZZZZZXXXXXXXXXXX<u>CATC</u>GGTGAGCCCGGGCTGTC**GGCGT**NN
TNNNANNTNNNANNTNNNANNTNNNANNTNNNANN**ATATG**

and

ZZZZZZZZZXXXXXXXXXXX<u>CCTC</u>TACGCCGGACGCATCGTGCGGCCGC
CCGGGC**CACCG**NNTNNNANNTNNNANNTNNNTNNTNNNANNTN
NNANN**AGCGT**

The Z's at the beginning of each read are random nucleotides used as unique molecular identifiers (UMIs) to correct for PCR jackpotting (Kivioja *et al,* 2012), the X's are the sample multiplexing tag sequences, and the N's are the random nucleotides of the DNA barcodes. To minimize the chances of barcode crosstalk, we used dual barcodes, with independent random barcode sequences on the forward and reverse reads and 27 random nucleotides in each of the forward and reverse barcodes.

The raw sequences were parsed, and sequences were kept for further analysis only if they passed the following quality criteria for both the forward and reverse reads:

- The four bases after the multiplexing tag (underlined in the sequences above) must match the nominal sequence with one allowed mismatch, and the multiplexing tag sequence (X's in the sequences above) must match the nominal sequence for one of the multiplexing tags used with up to three allowed mismatches.
- The five flanking bases before and after the barcodes (bold in the sequences above) must match the nominal sequence with one allowed mismatch per set of five bases, and the number of bases in the barcode must be between 35 and 41 (inclusive).
- The mean Illumina quality score for the barcode and the five flanking bases before and after the barcode must be greater than 30.

For the four lanes of HiSeq data, there were 2,024,537,456 raw reads, of which 1,576,168,836 reads passed the quality criteria (78%). Note that 20% of the DNA sample loaded onto the HiSeq instrument was phiX DNA.

True barcode sequences were identified using the Bartender1.1 clustering algorithm (Zhao *et al,* 2018) with the following parameter settings: maximum cluster distance = 4, cluster merging threshold = 8, cluster seed length = 5, cluster seed step = 1, and frequency cutoff = 500. Barcodes from the forward and reverse reads were clustered independently. The Bartender1.1 clustering algorithm identified 43,259 distinguishable forward barcode clusters and 31,055 distinguishable reverse barcode clusters.

To correct for insertion–deletion read errors, barcode clusters of different length were considered for merging. First, barcode clusters with sequences that were sub-strings of one another were automatically merged. Second, pairs of barcode clusters with a DNA sequence Levenshtein distance of 1 or 2 were merged if the ratio of the smaller cluster read count to the total read count of both clusters was < 0.001 and 0.0001, respectively. Third, all barcode clusters with a Levenshtein distance < 7 from the barcode for the spike-in control were merged.

After merging barcode clusters of different lengths, there were 43,169 distinguishable forward barcode clusters and 30,931 distinguishable reverse barcode clusters. The random positions within the forward and reverse barcodes had approximately equal probabilities for each nucleotide, with a mean entropy per position of 1.9799 bits ± 0.0066 bits.

After barcode clustering and merging, the barcode sequencing reads were sorted based on the sample multiplexing tags and the barcode read counts were corrected for PCR jackpotting effects. Sets of multiple barcode reads were treated as PCR jackpot duplicates if they had the same UMI sequence, the same multiplexing tag, and the same barcode sequence for both forward and reverse barcode reads. In the corrected barcode count, each set of PCR jackpot duplicates was counted as a single read. Approximately 15% of the total barcode sequencing reads were found to be PCR jackpot duplicates.

The forward and reverse barcodes were then combined to give the DNA barcodes used to measure the relative abundance of each LacI variant in the library. An additional barcode count threshold was applied, keeping only DNA barcodes with a total read count (across all 24 environments and 4 time points) greater than 2,000. A small number (139) of DNA barcodes were identified as likely chimeras with forward and reverse barcodes combined from different plasmid templates (Smyth *et al*, 2010; Schlecht *et al*, 2017; Omelina *et al*, 2019). The likely chimera barcodes were not used in further analysis.

Finally, 14 pairs of DNA barcodes were found with DNA sequence Hamming distance of one (across both forward and reverse barcodes). Only one DNA barcode from each pair was also found in the long-read sequencing data (see Long-read sequencing section, below). In addition, the fitness curves (vs IPTG concentration) were very similar for both barcodes in each pair. Based on this, the read counts associated with each of those 14 pairs of dual barcodes were merged, and each pair was treated as a single DNA barcode.

The final set of 67,730 DNA barcodes was used for all subsequent analysis to extract estimates of the fitness and dose-response curve associated with each barcode.

### Long-read sequencing

The full sequence of the library plasmid (pTY1) for every LacI variant in the library was measured using PacBio circular consensus HiFi sequencing. The stock of *E. coli* containing the library was grown in media and plasmid was purified by miniprep. Purified plasmid DNA was linearized with BspOI restriction enzyme digest and submitted for sequencing (University of Maryland Institute of Genome Sciences). The HiFi sequencing data were used to determine the consensus *lacI* sequence for each variant and the corresponding DNA barcode. Of the 67,731 distinct DNA barcodes (see Barcode sequencing section, above), the HiFi sequencing data were

used to determine the *lacI* sequences for 63,064 (93%), 3,878 with a single HiFi lacI read, and 59,186 with multiple HiFi lacI reads.

In addition, the full plasmid sequence was used to detect unintended mutations in the plasmid, i.e., mutations to plasmid regions other than the *lacI* CDS. For analysis of the HiFi read data, the full plasmid sequence was divided into 11 non-overlapping regions that roughly correspond to different functional elements of the plasmid (Appendix Table S6), and the sequences for each region were extracted from the HiFi reads using a custom bioinformatic pipeline (https://github.com/djross22/nist_lacI_landscape_analysis). The number of unintended mutations to plasmid regions other than the *lacI* CDS was relatively low (Appendix Table S6), so it was not possible to examine mutational effects with base pair- or residue-level resolution. However, by pooling the mutational information for each region, significant region-specific effects could be detected. To determine if mutations in a region of the plasmid had a significant effect, the estimated Hill equation parameters were compared for all variants with one or more mutations in a given plasmid region vs all variants with zero mutations in that region. Significant differences in the geometric mean of one or more Hill equation parameters were found for variants with mutations in the following regions: tetA (*P*-value for $\log_{10}(G_\infty)$: $2 \times 10^{-56}$), KAN (*P*-value for $\log_{10}(G_\infty)$: $4 \times 10^{-11}$), origin of replication (*P*-value for $\log_{10}(G_\infty)$: $6 \times 10^{-14}$), and YFP (*P*-value for $\log_{10}(G_0)$: $4 \times 10^{-109}$; *P*-value for $\log_{10}(G_\infty)$: $5 \times 10^{-10}$; *P*-value for $\log_{10}(EC_{50})$: $2 \times 10^{-74}$), where the *P*-values given are for Welch's unequal-variances *t*-test.

In addition, 43 of the 535 variants with the wild-type LacI amino acid sequence had mutations in the regulatory region (containing the $P_{lacI}$ and $P_{tacI}$ promoters, the *lacO* operator, the riboJ insulator, and the RBS sites for both *lacI* and *tetA*). Of those 43 variants, three had $EC_{50}$ values that differed by approximately 2-fold or more from the geometric mean value for the wild-type $EC_{50}$. The Kolmogorov-Smirnov test was used to compare the distributions of $EC_{50}$ values between the wild-type variants with and without mutations in the regulatory region; the results indicated a significant difference (*P*-value: 0.024).

To avoid biasing the results of the machine learning and other quantitative phenotypic analyses, variants were excluded from those analyses if they had one or more mutations in the non-*lacI* regions that show significant mutational effects: tetA, YFP, KAN, the origin of replication, and the regulatory region. After applying this data quality filter in addition to those described above, there were 54,162 variants that we used for further quantitative analysis.

### Library-scale fitness measurement

The experimental approach for this work was designed to maintain bacterial cultures in exponential growth phase for the full duration of the measurements. So, in all analyses, the Malthusian definition of fitness was used, i.e., fitness is the exponential growth rate (Wu *et al*, 2013).

The fitness of cells containing each LacI variant was calculated from the change in the relative abundance of DNA barcodes over time. The spike-in control was used to normalize the DNA barcode count data to enable the determination of the absolute fitness for each LacI variant in the library.

Briefly, for each LacI variant in each of the 24 chemical environments, the ratio of the barcode read count to the spike-in read count was fit to a function assuming exponential growth and a delay in

the onset of the fitness impact of tetracycline. The fitness associated with each variant in each of the 24 chemical environments was determined as a parameter in the corresponding least-squares fit as detailed below.

The barcode sequencing data were analyzed with a model based on the assumption that the number of cells containing each LacI variant grows with an exponential expansion rate that is independent of all other variants. So, for each sample, at the end of the incubation cycle for Growth Plate $j$, the number of cells with LacI variant $i$ is:

$$N_{i,j} = \frac{N_{i,j-1}}{d}\exp(\mu_{i,j}\Delta t) \tag{1}$$

where, $d$ ($= 10$) is the dilution factor used in transferring the cell culture from Growth Plate $j-1$ to Growth Plate $j$, $\Delta t$ ($\approx 165$ min) is the total incubation time for each growth plate (including time required for automated cell passaging), and $\mu_{i,j}$ is the fitness (i.e. mean exponential growth rate) of cells with LacI variant $i$ in Growth Plate $j$. Note that each growth plate was pre-heated to 37°C before transferring cells from the previous growth plate, so the cell growth rate was assumed to be unaffected by temperature changes during passaging.

For samples without tetracycline, the chemical composition of the media was the same for all growth plates, so the fitness is assumed to be constant, $\mu_{i,j} = \mu_i^0$, where $\mu_i^0$ is the fitness associated with LacI variant $i$ in the absence of tetracycline. Consequently, the number of cells in each Growth Plate for samples grown without tetracycline is:

$$\log\left(N_{i,j}^0\right) = \log(N_{i,0}^0) + j(\mu_i^0\Delta t - \log(d)) \tag{2}$$

where $N_{i,j}^0$ is the number of cells with LacI variant $i$ at the end of Growth Plate $j$ for samples grown without tetracycline.

For samples grown with tetracycline, the tetracycline was only added to the culture media for Growth Plates 2–4. Because of the mode of action of tetracycline (inhibition of translation), there was a delay in its effect on cell fitness: Immediately after diluting cells into Growth Plate 2 (the first plate with tetracycline), the cells still had a normal level of proteins needed for growth and proliferation and they continued to grow at nearly the same rate as without tetracycline. Over time, as the level of proteins required for cell growth decreased due to tetracycline, the growth rate of the cells decreased. Accordingly, the analysis accounts for the variation in cell fitness (growth rate) as a function of time after the cells were exposed to tetracycline. With the assumption that the fitness is approximately proportional to the number of proteins needed for growth, the fitness as a function of time is taken to approach the new value with an exponential decay:

$$\mu_{i,j} = \mu_i^0 + \left(\mu_i^{tet} - \mu_i^0\right)e^{-\alpha j} \tag{3}$$

where $\mu_i^{tet}$ is the steady-state fitness with tetracycline, and $\alpha$ is a transition rate. The transition rate was kept fixed at $\alpha = \log(5)$, determined from a small-scale calibration measurement. Note that at the tetracycline concentration used during the library-scale measurement (20 µg/ml), $\mu_i^{tet}$ was greater than zero even at the lowest $G(L)$ levels (Appendix Fig S10). From equation (3), the number of cells in each Growth Plate for samples grown with tetracycline is:

$$\log\left(N_{i,j}^{tet}\right) = \log(N_{i,0}^{tet}) + j(\mu_i^{tet}\Delta t - \log(d)) + \frac{\Delta t}{\alpha}\left(\mu_i^0 - \mu_i^{tet} + \left(\mu_i^{tet} - \mu_i^0\right)e^{-\alpha j}\right). \tag{4}$$

The barcode read count for variant $i$ in Growth Plate $j$ was assumed to be proportional to the cell number:

$$R_{i,j} = a_i b_i N_{i,j} \tag{5}$$

where $a_i$ is a proportionality constant associated with variant $i$, and $b_j$ is a proportionality constant associated with Growth Plate $j$. The proportionality constant $a_i$ can be different for each variant $i$ due to differences in PCR amplification efficiency resulting from variations in the barcode sequences on each amplicon. Similarly, the proportionality constant $b_j$ can be different for each Growth Plate because of sample-to-sample variations in the DNA extraction efficiency or differences in PCR efficiency associated with different sample multiplexing tag sequences.

The logarithm of the read count normalized by the spike-in read count was used to estimate the fitness of each variant from its associated barcode read count:

$$\log(r_{i,j}) \equiv \log\left(\frac{R_{i,j}}{R_{spike,j}}\right). \tag{6}$$

For samples without tetracycline, $\mu_i^0$ was estimated for each variant using a weighted linear least-squares fit to the log-count ratio vs $j$:

$$\log\left(r_{i,j}^0\right) = \log(r_{i,0}^0) + j\Delta\mu_i^0\Delta t \tag{7}$$

where $r_{i,j}^0 \equiv \frac{a_i}{a_{spike}}\frac{N_{i,0}^0}{N_{spike,0}^0}$, and $\Delta\mu_i^0 \equiv \mu_i^0 - \mu_{spike}^0$ is the difference between the fitness of variant $i$ and the spike-in fitness without tetracycline.

For samples grown with tetracycline, $\mu_i^{tet}$ was estimated for each variant with a weighted least-squares fit to the nonlinear form for the log-count ratio:

$$\log\left(r_{i,j}^{tet}\right) = \log(r_{i,0}^{tet}) + j\Delta\mu_i^{tet}\Delta t + \frac{\Delta t}{\alpha}\left(\Delta\mu_i^0 - \Delta\mu_i^{tet} + \left(\Delta\mu_i^{tet} - \Delta\mu_i^0\right)e^{-\alpha j}\right) \tag{8}$$

where $r_{i,j}^{tet} \equiv \frac{a_i}{a_{spike}}\frac{N_{i,0}^{tet}}{N_{spike,0}^{tet}}$, and $\Delta\mu_i^{tet} \equiv \mu_i^{tet} - \mu_{spike}^{tet}$ is the difference between the fitness of variant $i$ and the spike-in fitness with tetracycline.

For the least-squares fits to determine both $\mu_i^0$ and $\mu_i^{tet}$, the fits were weighted based on the propagated uncertainties of $r_{i,j}^0$ and $r_{i,j}^{tet}$ calculated assuming that the uncertainty of each read count was dominated by Poisson sampling.

For the fitness landscape measurement, there were a large number of outliers for the read count measurements from three of the samples: Growth Plate 3, without tetracycline, [IPTG] = 8 µmol/l; Growth Plate 4, without tetracycline, [IPTG] = 64 µmol/l and [IPTG] = 2,048 µmol/l. These three samples were excluded from the analysis.

### Dose-response curve measurements

Plasmids pTY1 and pVER were engineered to provide two independent measurements of the dose-response curve for LacI variants. First, in pTY1, LacI regulates the expression of a

tetracycline resistance gene (*tetA*) that enables determination of the dose-response from barcode sequencing data by comparing the fitness measured with tetracycline to the fitness measured without tetracycline. Second, in pVER, LacI regulates the expression of a fluorescent protein (YFP) that enables direct measurement of the dose-response curve with flow cytometry.

A set of nine randomly selected LacI variants were used to calibrate the estimation of regulated gene expression output from the barcode sequencing fitness measurements (Appendix Fig S10). The calibration data consisted of the fitness data for each calibration variant from the library barcode sequencing measurement (using the library plasmid, pTY1) and flow cytometry data for each calibration variant prepared as a clonal culture (using the verification plasmid, pVER). These data were fit to a Hill equation model for the fitness impact of tetracycline as a function of the regulated gene expression level, $G$:

$$\frac{\mu^{tet}}{\mu^0} - 1 = \Delta f\left(\frac{G^{n_f}}{G_{50}^{n_f} + G^{n_f}} - 1\right) \tag{9}$$

where $\mu^{tet}$ is the fitness with tetracycline, $\mu^0$ is the fitness without tetracycline, $\Delta f$ is the maximal fitness impact of tetracycline (when $G = 0$), $G_{50}$ is the gene expression level that produces a 50% recovery in fitness, and $n_f$ characterizes the steepness of the fitness calibration curve. Because the fitness calibration curve, equation (9), is nonlinear, it cannot be directly inverted to give the regulated gene expression level for all possible fitness measurements. So, two Bayesian inference models were used to estimate the dose-response curves for every LacI variant in the library using the barcode sequencing fitness measurements. Source code for both models is included in the software archive at https://github.com/djross22/nist_lacI_landscape_analysis. Both inference models used equation (9) to represent the relationship between fitness and regulated gene expression. The parameters $\Delta f$, $G_{50}$, and $n_f$ were included in both inference models as parameters with informative priors. Priors for $G_{50}$ and $n_f$ were based on the results of the fit to the fitness calibration data (Appendix Fig S10: $G_{50} \sim$ normal (mean = 13,330, SD = 500), $n_f \sim$ normal(mean = 3.24, SD = 0.29). We chose the prior for $\Delta f$ based on an examination of $\mu^{tet}/\mu^0 - 1$ measured with zero IPTG: $\Delta f \sim$ exponentially-modified-normal (mean = 0.720, SD = 0.015, rate = 14). The use of a prior for $\Delta f$ with a broad right-side tail was important to accommodate variants in the library for which $\mu^{tet}/\mu^0 - 1$ was systematically less than −0.722.

The first Bayesian inference model assumed that the dose-response curve for each LacI variant was described by the Hill equation. The Hill equation parameters for each variant, $G_\infty$, $G_0$, $EC_{50}$, and $n$ and their associated uncertainties were determined using Bayesian parameter estimation by Markov Chain Monte Carlo (MCMC) sampling with PyStan (Carpenter *et al*, 2017). Broad, flat priors were used for $\log_{10}(G_0)$, $\log_{10}(G_\infty)$, and $\log_{10}(EC_{50})$, with error function boundaries to constrain those parameter estimates to within the measurable range (100 MEF $\leq G_0$, $G_\infty \leq$ 50,000 MEF; 0.1 μmol/l $\leq EC_{50,i} \leq$ 40,000 μmol/l). The prior for $n_i$ was a gamma distribution with shape parameter of 4.0 and inverse scale parameter of 3.33. The inference model was run individually for each LacI variant, with four independent chains, 1,000 iterations per chain

(500 warmup iterations), and the adapt_delta parameter set to 0.9. Testing with data from a set of randomly selected variants indicated that these settings for the Stan sampling algorithm typically produced a Gelman-Rubin $\hat{R}$ diagnostic < 1.05 and number of effective iterations > 100.

The second Bayesian inference model was a non-parametric GP model (Rasmussen & Williams, 2005) that assumed only that the dose-response curve for each LacI variant was a smooth function of IPTG concentration. The GP model was used to determine which variants had band-pass or band-stop phenotypes. The GP model was also implemented using MCMC sampling with PyStan (Carpenter *et al*, 2017). The GP inference model was run individually for each variant, with four independent chains, 1,000 iterations per chain (500 warmup iterations), and the adapt_delta parameter set to 0.9. Testing with data from a set of randomly selected variants indicated that these settings for the Stan sampling algorithm of the GP model typically produced a Gelman-Rubin $\hat{R}$ diagnostic less than 1.02 and number of effective iterations greater than 200.

### Flow cytometry measurements

Over 100 LacI variants from the library were chosen for flow cytometry verification of the dose-response curves. The CDSs of these variants were chemically synthesized (Twist Bioscience). Each synthesized sequence was digested with restriction enzymes XhoI and SgsI, and ligated into the verification plasmid, pVER, and then transformed into MG1655$\Delta$*lac*. Transformants were plated on LB agar supplemented with kanamycin and 0.2% glucose. LacI variant sequences were verified with Sanger sequencing (Psomagen USA). For flow cytometry measurements of dose-response curves, a culture of *E. coli* containing pVER with a chosen variant sequence was distributed across 12 wells of a 96-well plate and grown to stationary phase using the automated microbial growth system. After growth to stationary phase, cultures were diluted 50-fold into a plate containing the same 12 IPTG concentrations used during the fitness landscape measurement (0–2,048 μmol/l). In some cases, higher IPTG concentrations were used to capture the full dose-response curves of selected variants (e.g., Appendix Figs S5–S8). Cultures were then grown for 160 min (~ 3.3 generations) before being diluted 10-fold into the same IPTG gradient and grown for another 160 min. Then, 5 μl of each culture was diluted into 195 μl of PBS supplemented with 170 μg/ml chloramphenicol and incubated at room temperature for 30–60 min to halt the translation of YFP and allow extant YFP to mature in the cells.

Samples were measured on an Attune NxT flow cytometry with autosampler using a 488 nm excitation laser and a 530 nm ± 15 nm band-pass emission filter. Blank samples were measured with each batch of cell measurements, and an automated gating algorithm was used to discriminate cell events from non-cell events (Appendix Fig S27A and B). With the Attune cytometer, the area and height parameters for each detection channel are calibrated to give the same value for singlet events. So, to identify singlet cell events and exclude multiplet cell events, a second automated gating algorithm was applied to select only cells with side scatter area $\cong$ side scatter height (Appendix Fig S27C and D). All subsequent analysis was performed using the singlet cell event data. Fluorescence data were calibrated to molecules of equivalent fluorophore (MEF) using fluorescent calibration beads (Spherotech,

part no. RCP-30-20A). The cytometer was programmed to measure a 25 µl portion of each cell sample, and the 40-fold dilution used in the cytometry sample preparation resulted in approximately 20,000 singlet cell measurements per sample. The geometric mean of the YFP fluorescence was used as a summary statistic to represent the regulated gene expression level as a function of the input ligand concentration, [IPTG] for each LacI variant. An autofluorescence control (strain MG1655Δ*lac* with a plasmid similar to pVER but lacking the YFP gene) was also measured with flow cytometry and analyzed in the same way as other variants. The regulated gene expression output level for each variant is reported as the geometric mean of the measured fluorescence for that variant minus the geometric mean of the measured fluorescence for the zero-fluorescence control (92 MEF).

Hill equation parameters were estimated from the flow cytometry data using Bayesian parameter estimation by Markov Chain Monte Carlo (MCMC) sampling with PyStan (Carpenter *et al*, 2017). The Bayesian inference model used for flow cytometry data analysis assumed that the flow cytometry data resulted from a Hill equation response plus normally distributed measurement errors. The model used the same priors for the Hill equation parameters as described above.

### Calculation of abundance for LacI phenotypes

The relative abundance of the various LacI phenotypes (Appendix Fig S3) was estimated using the results of both Bayesian inference models (Hill equation and GP). Variants were labeled as "flat response" if the Hill equation model and the GP model agreed (i.e., if the median estimate for the Hill equation dose-response curve was within the central 90% credible interval from the GP model at all 12 IPTG concentrations) and if the posterior probability for $G_0 > G_\infty$ was between 0.05 and 0.95 (from the Hill equation model inference). Variants were labeled as having a negative response if the slope, $\partial G/\partial L$, was negative at one or more IPTG concentrations with 0.95 or higher posterior probability (from the GP model inference). To avoid false positives from end effects, this negative slope criteria was only applied for IPTG concentrations between 2 µmol/l IPTG and 1,024 µmol/l. Variants were labeled as "always on" (the $\mathit{\Gamma}$ phenotype from reference (Markiewicz *et al*, 1994)) if they were flat-response and if $G(0)$ was greater than 0.25 times the wild-type $G_\infty$ value with 0.95 or higher posterior probability (from the GP model inference). Variants were labeled as "always off" (the $I^S$ phenotype from reference (Markiewicz *et al*, 1994)) if they were flat-response but not always on. Variants were labeled as band-stop or band-pass if the slope, $\partial G/\partial L$, was negative at some IPTG concentrations and positive at other IPTG concentrations, both with 0.95 or higher posterior probability (from the GP model inference). Band-stop and band-pass variants were distinguished by the ordering of the negative-slope and positive-slope portions of the dose-response curves. Variants that had a negative response but that were not band-pass or band-stop, were labeled as inverted. False-positive rates were estimated for each phenotypic category by manually examining the fitness vs IPTG data for LacI variants with less than three substitutions. Typical causes of false-positive phenotypic labeling included unusually high noise in the fitness measurement and biased fit results due to outlier fitness data points. Estimated false-positive rates ranged between 0.001

and 0.005. The relative abundance values shown in Appendix Fig S3A were corrected for false positives using the estimated rates.

### Comparison of synonymous mutations

The library contained a set of 39 variants with the wild-type *lacI* CDS (each with a different DNA barcode), and a set of 310 variants with only synonymous nucleotide changes (i.e., no amino acid substitutions). Both sets had long-read sequencing coverage for the entire plasmid and were screened to retain only variants with zero unintended mutations in the plasmid (i.e., no mutations in regions of the plasmid other than the *lacI* CDS). The Hill equation fit results for those two sets were compared to determine whether synonymous nucleotide changes significantly affected the phenotype. The Kolmogorov-Smirnov test was used to compare the distributions of Hill equation parameters between these two sets. The resulting *P*-values (0.71, 0.40, 0.28, and 0.17 for $G_0$, $G_\infty$, $EC_{50}$, and $n$, respectively) indicate that there were no significant differences between them. Additionally, the library contained 40 sets of variants, each with four or more synonymous CDSs (including the set of synonymous wild-type sequences and 39 non-wild-type sequences). A hierarchical model was used to compare the Hill equation parameters within each set of synonymous CDSs. Within each set, the uncertainty associated with individual variants was typically larger than the variant-to-variant variability estimated by the hierarchical model. Overall, these results indicate that synonymous SNPs did not measurably impact the LacI phenotype, so only the amino acid sequences were considered for any subsequent quantitative genotype-to-phenotype analysis.

### Analysis of single-substitution data

The single amino acid substitution results presented in Figs 3 and, 6B and C, Appendix Figs S17–S20, and included in Dataset EV1 are a combination of direct experimental observations, DNN model results, and estimates of $G_0$ for missing substitutions.

For direct experimental observations, multiple LacI variants were often present in the library with the same single substitution. To ensure that the highest quality data was used for the single-substitution analysis, only data for variants with more than 5,000 total barcode reads were used (see Barcode sequencing section, above). For each single substitution, if there was only one LacI variant with more than 5,000 barcode reads, the median and standard deviation for each parameter were used directly from the Bayesian inference using the Hill equation model. If there was more than one LacI variant with a given single substitution and more than 5,000 barcode reads, the consensus Hill equation parameter values and standard deviations for that substitution were calculated using a hierarchical model based on the eight schools model (Rubin, 1981; https://mc-stan.org/users/documentation/case-studies/divergences_and_bias.html). The hierarchical model was applied separately for each Hill equation parameter. The logarithm of the parameter values was used as input to the hierarchical model, and the input data were centered and normalized by 1.15 × the minimum measurement uncertainty. The standard normal distribution was used as a loosely informative prior for the consensus mean effect, and a half-normal prior (mean = 0.5, SD = 1) was used for the normalized consensus standard deviation (i.e., hierarchical standard deviation). These priors and normalization were chosen so that the model gave intuitively reasonable results for the consensus of two LacI variants (i.e., close

to the results for the LacI variant with the lowest measurement uncertainty). Results for the hierarchical model were determined using Bayesian parameter estimation by Markov Chain Monte Carlo (MCMC) sampling with PyStan (Carpenter *et al*, 2017). MCMC sampling was run with four independent chains, 10,000 iterations per chain (5,000 warmup iterations), and the adapt_delta parameter set to 0.975.

For $G_0$, the direct experimental results were used for the 1,047 substitutions plotted as gray points or red points and error bars in Fig 3D and Appendix Fig S20. In addition, estimated values were used for the 83 missing substitutions that have been previously shown to result in an "always on" LacI phenotype (i.e., the $I^-$ phenotype (Markiewicz *et al*, 1994; Pace *et al*, 1997)). For these substitutions, plotted as pink-gray points and error bars in Fig 3D, the median value was estimated to be equal to the wild-type value for $G_\infty$ (24,000 MEF), and the geometric standard deviation was estimated to be 4-fold, both based on information from previous publications (Markiewicz *et al*, 1994; Pace *et al*, 1997). Note that these 83 substitutions are completely missing from the experimental landscape dataset reported here, i.e., they are not found in any LacI variant, as single substitutions or in combination with other substitutions.

For $G_\infty$ and $EC_{50}$, the direct experimental results were used for the 964 substitutions that are found as single substitutions in the library and that have a consensus standard deviation for $\log_{10}(EC_{50})$ less than 0.35. An additional 74 substitutions are found as single substitutions in the library, but with higher $EC_{50}$ uncertainty. For these substitutions, either $EC_{50}$ is comparable to or higher than the maximum ligand concentration used for the measurement (2,048 μmol/l IPTG), or $G_\infty$ is comparable to $G_0$ (or both). Consequently, the dose-response curve is flat or nearly flat across the range of concentrations used, and the Bayesian inference used to estimate the Hill equation parameters results in $EC_{50}$ and $G_\infty$ estimates with large uncertainties. The DNN model can provide a better parameter estimate for these flat-response variants because it uses data and relationships from the full library (e.g., the log-additivity of $EC_{50}$) to predict parameter values for each single substitution. So, the DNN model results were used for these 74 substitutions. Finally, the DNN model results were used for an additional 953 substitutions that are found in the library, but only in combination with other substitutions (i.e., not as single substitutions).

### Identification of high-frequency substitutions and structural features associated with inverted and band-stop phenotypes

The set of 43 strongly inverted LacI variants discussed in the main text and used for the plots in Fig 5A,C were identified by the following criteria: $G_0/G_\infty \geq 2$, $G_0 > G_{\infty,wt}/2$, $G_\infty < G_{\infty,wt}/2$, and $EC_{50}$ between 3 μmol/l and 1,000 μmol/l. The set of 31 strong band-stop variants discussed in the main text and used for the plots in Fig 5B and D were identified by the following criteria: $G_0 > G_{\infty,wt}/2$, $G_{min} < G_{\infty,wt}/2$, and the slope, $\partial\log(G)/\partial\log(L)$, of less than $-0.07$ at low IPTG concentrations and greater than zero at higher IPTG concentrations, both with 0.95 or higher posterior probability (from the GP model inference). To avoid false positives due to noise in the DNA barcode counting, only LacI variants with a total DNA barcode read count greater than 3,000 were included. Also, the sets of strongly inverted and strong band-stop variants were manually screened for additional likely false positives due to outlier fitness data points.

A hypergeometric test was used to determine the amino acid substitutions that occur more frequently in the set of strongly inverted or strong band-stop variants than in the full library (the set of 52,321 variants with more than 3,000 DNA barcode reads and with the *lacI* sequences determined by long-read sequencing). For each possible substitution, the cumulative hypergeometric distribution was used to calculate the probability of the observed number of occurrences of that substitution in the set of inverted or band-stop variants under a null model of no association. This probability was used as a *P*-value for the null hypothesis that the observed number of inverted or band-stop variants with that substitution resulted from an unbiased random selection of variants from the full library. Substitutions were considered to occur at significantly higher frequency if they had a *P*-value $< 0.005$ and if they occurred more than once in the set of inverted or band-stop variants. In the set of strongly inverted variants, 10 associated (higher frequency) amino acid substitutions were identified: S70I, K84N, D88Y, V96E, A135T, V192A, G200S, Q248H, Y273H, and A343G. In the set of strong band-stop variants, eight associated substitutions were identified: V4A, A92V, H179Q, R195H, G178D, G265D, D292G, and R351G. To estimate the number of false positives, random sets of LacI variants were chosen with the same sample size as the strongly inverted (43) or the strong band-stop (31) variants and the same significance criteria was applied. From 300 independent iterations of the random selection, the estimated mean number of false-positive substitutions was 2.1 and 2.3 for the inverted and band-stop phenotypes, respectively. Statistics for the test results are given in Appendix Tables S3 and S4.

A similar procedure was used to determine which structural features within the protein are mutated with higher frequency in the inverted or band-stop LacI variants. The structural features considered were the secondary structures from the complete crystal structure of LacI (Lewis *et al*, 1996), as well as larger structural features (N-terminal core domain, C-terminal core domain, DNA-binding domain, dimer interface) and functional domains (ligand-binding, core-pivot). All of the domains and features included in the analysis are listed in Appendix Table S2. The *P*-value threshold used for significance was 0.025. For the strongly inverted variants, five features were identified with a higher frequency of amino acid substitutions: the dimer interface, residues within 7 Å of the ligand-binding pocket, helix 5, helix 11, and β-strand I. For the strong band-stop variants, three features were identified: the C-terminal core, β-strand J, and helix 9. From 300 independent random selections of variants from the full library, the estimated mean number of false-positive features was 0.38 and 0.51 for the inverted and band-stop phenotypes, respectively. Statistics for the test results are given in Appendix Tables S3 and S4.

### Deep neural network modeling

The dataset was pruned to a set of high-quality sequences for DNN modeling. Specifically, data for a LacI variant was only used for modeling if it satisfied the following criteria:

- No mutations were found in the long-read sequencing results for the regions of the plasmid encoding kanamycin resistance, the origin of replication, the tetA and YFP genes, and the regulatory region containing the promoters and ribosomal binding sites for *lacI* and tetA (Appendix Table S6).

- The total number of barcode read counts for a LacI variant was > 3,000.
- The number of amino acid substitutions was < 14.
- The measurement uncertainty for $\log_{10}(G_\infty)$ was < 0.7.
- The results of the Hill equation model and the GP model agreed at all 12 IPTG concentrations. More specifically, data were only used if the median estimate for the dose-response curve from the Hill equation model was within the central 90% credible interval from the GP model at all 12 IPTG concentrations.

After applying the quality criteria listed above, 47,462 LacI variants remained for DNN modeling. The data were used to train the DNN model to predict the Hill equation parameters $G_0$, $G_\infty$, and $EC_{50}$ as detailed below.

Amino acid sequences were represented as one-hot encoded vectors of length $L = 2,536$, and with mutational paths represented as $K \times L$ tensors for a sequence with K substitutions. The logarithm of the Hill equation parameter values were normalized to a standard deviation of 1, and then shifted by the corresponding value of the wild-type sequence in order to correctly represent the prediction goal of the change in each parameter relative to wild-type LacI. A long-term, short-term recurrent neural network was selected for the underlying model (Hochreiter & Schmidhuber, 1997), with 16 hidden units, a single hidden layer, and hyperbolic tangent (tanh) non-linearities. Inference was performed in pytorch (preprint: Paszke *et al*, 2019) using the Adam optimizer (preprint: Kingma & Ba, 2017). For $EC_{50}$ and $G_0$, the contribution of individual data points to the regression loss were weighted inversely proportional to their experimental uncertainty. Model selection was performed with 10-fold cross-validation on the training set (80% of all available data). Approximate Bayesian inference was performed with the Bayes-by-backprop approach (preprint: Blundell *et al*, 2015). Briefly, this substitutes the point-estimate parameters of the neural network with variational approximations to a Bayesian model, represented as a mean and variance of a normal random variable. Effectively, this only doubles the number of parameters in the model. A mixture of two normal distributions was used as a prior for each parameter weight, with the two mixture components having high and low variance, respectively. This prior emulates a sparsifying spike-slab prior while remaining tractable for inference based on back propagation. Posterior means of each weight were used to calculate posterior predictive means, while Monte Carlo draws from the variational posterior were used to calculate the model prediction uncertainty (Appendix Fig S14).

Variational approximations typically underestimate uncertainty. So, to correct the uncertainty estimates, the model prediction uncertainty obtained from the variational approximation was compared with the model root-mean-square error (RMSE; i.e., the root-mean-square difference between the model prediction and the experimental measurement). For all three Hill equation parameters ($G_0$, $G_\infty$, and $EC_{50}$), both the prediction uncertainty and the RMSE increase with the number of amino acid substitutions relative to wild-type sequence (Appendix Fig S14A and B), and the RMSE at each substitutional distance is an approximately linear function of the median model uncertainty (Appendix Fig S13C). So, for the single-substitution analysis (Figs 3 and, 6B and C, Appendix Fig S20, Dataset EV1), the uncertainties from the variational approximation were multiplied by a factor of 3.8. This rescaled the uncertainties so that

the median uncertainty was approximately equal to the RMSE for each substitutional distance.

We compared the performance of the recurrent DNN model against two alternative models (Appendix Fig S12B). First, we trained a linear-additive model, which assumes that each parameter is log-additive and so only learns the average effect of each amino substitution across the entire dataset. Second, we trained a more conventional feed-forward neural network. The feed-forward neural network had four hidden layers, with 32, 64, 64, and 32 hidden units, respectively. Each hidden layer had a rectified linear unit (ReLU) non-linearity and a batch-normalization step between each layer. Both the linear-additive and feed-forward models were constructed and trained with pytorch (preprint: Paszke *et al*, 2019), using the Adam optimizer (preprint: Kingma & Ba, 2017) and a learning rate of $10^{-3}$.

# Data availability

Long-read and short-read DNA sequencing: NCBI BioProject PRJNA643436 (https://www.ncbi.nlm.nih.gov/bioproject/PRJNA643436).

pTY1 plasmid sequence: NCBI GenBank MT702633 (https://www.ncbi.nlm.nih.gov/nuccore/MT702633).

pVER plasmid sequence: NCBI GenBank MT702634 (https://www.ncbi.nlm.nih.gov/nuccore/MT702634).

The processed data table containing comprehensive data and information for each LacI variant in the library is publicly available via the NIST Science Data Portal, with the identifier ark:/88434/mds2-2259 (https://data.nist.gov/od/id/mds2-2259 or https://doi.org/10.18434/M32259). The data table includes the DNA barcode sequences, the barcode read counts for each sample and time point used for the library-scale measurement, fitness estimates for each barcoded variant across the 24 chemical environments, the results of both Bayesian inference models (including posterior medians, covariances, and 0.05, 0.25, 0.75, and 0.95 posterior quantiles), the LacI CDS and amino acid sequence for each barcoded variant (as determined by long-read sequencing), the number of LacI CDS reads in the long-read sequencing dataset for each barcoded variant, and the number of unintended mutations in other regions of the plasmid (from the long-read sequencing data).

All data analysis code is available at https://github.com/djross22/nist_lacI_landscape_analysis.

**Expanded View** for this article is available online.

### Acknowledgements
We would like to thank Vanya Paralanov, Daniel Samarov, Ben Scott, Zvi Kelman, Gilad Kusne, and Swarnavo Sarkar for thoughtful discussions during planning and execution of this work. We would also like to thank Jayan Rammohan, William Brad O'Dell, and Elizabeth Strychalski for insights during the experimental work, as well as improving the manuscript.

### Author contributions
DST, and DR conceived of the process. DST, SFL, and DR developed the experimental workflow. DST designed, built, and tested genetic constructs. EFR, and DR programmed automated protocols. DST, EFR, NA, OV, and DR performed landscape and verification experiments. PDT and DR performed Bayesian inference and model fitting. PDT designed and evaluated the recurrent architecture

for machine learning. PDT, NDO, and DR contributed to long-read sequencing analysis. DST, PDT, AP, and DR wrote the manuscript. All authors contributed to the manuscript.

## Conflict of interest

The authors declare that they have no conflict of interest. Certain commercial equipment, instruments, or materials are identified to adequately specify experimental procedures. Such identification neither implies recommendation nor endorsement by the National Institute of Standards and Technology nor that the equipment, instruments, or materials identified are necessarily the best for the purpose.

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
