## [Review Process File · Molecular Systems Biology]

The genotype-phenotype landscape of an allosteric protein

David Ross, Drew Tack, Peter Tonner, Abe Pressman, Nathanael Olson, Sasha Levy, Eugenia Romantseva, Nina Alperovich, and Olga Vailjeva

DOI: [10.15252/msb.202010179](https://doi.org/10.15252/msb.202010179)

Corresponding author(s): David Ross (david.ross@nist.gov)

Review Timeline:	Transfer from Review Commons :	14th Dec 20
	Editorial Decision:	16th Dec 20
	Revision Received:	23rd Dec 20
	Editorial Decision:	5th Feb 21
	Revision Received:	15th Feb 21
	Accepted:	18th Feb 21

Editor: Maria Polychronidou

Transaction Report: *This manuscript was transferred to Molecular Systems Biology following peer review at Review Commons*

**Review
COMMONS**

Thank you for submitting your manuscript to Molecular Systems Biology. I have now read the manuscript and your point-by-point response to the comments of the Review Commons reviewers. I would like to invite you to submit your revised manuscript to Molecular Systems Biology once you have completed the pending revisions.

Overall, we agree with the reviewers that the presented analyses and dataset seem relevant. I think that the performed revisions in combination with your plan to compare the performance of the deep neural network model to that of additive linear model, sound promising for addressing the reviewers' concerns.

The eventual acceptance of the study will depend on how well the issues raised by the referees have been addressed. The revised study may be re-reviewed (by the original reviewers). As you might already know, our editorial policy allows in principle a single round of major revision, and therefore it is essential to provide responses to the reviewers' comments that are as complete as possible.

To speed up the evaluation of you revised manuscript we would also ask you to address the following editorial points.

Reviewer #1

The authors study allostery with a beautiful genotype-phenotype experiment to study the fitness landscape of an allosteric lac repressor protein. The authors make a mutational library using error prone pcr and measure the impact on antibiotic resistance protein expression at varying levels of ligand, IPTG, expression. After measuring the impact of mutations authors fill-in the missing data using a neural net model. This type of dose response is not standard in the field, but the richness of their data and the discovery of the "band pass" phenomena prove its worth here splendidly.

Using this mixed experimental/predicted data the authors explore how each mutation alters the different parameters of a hill equation fit of a dose response curve. Using higher order mutational space the authors look at how mutations can qualitatively switch phenotypes to inverted or band-stop dose-response curves. To validate and further explore a band-stop novel phenotype, the authors focused on a triple mutant and made all combinations of the 3 mutations. The authors find that only one mutation alone alters the dose-response and only in combination does a band-stop behavior present itself. Overall this paper is a fantastic data heavy dive into the allosteric fitness landscape of protein.

Overall, the data presented in this paper is thoroughly collected and analyzed making the conclusions well-based. We do not think additional experiments nor substantial changes are needed apart from including basic experimental details and more biophysical rationale/speculation as discussed in further detail below.

The authors do a genotype-phenotype experiment that requires extensive deep sequencing experiments. However, right now quite a bit of basic statistics on the sequencing is missing. Baseline library quality is somewhat shown in supplementary fig 2 but the figure is hard to interpret. It would be good to have a table that states how many of all possible mutations at different mutation depths (single, double, etc) there are. Similarly, sequencing statistics are missing- it would be useful to know how many reads were acquired and how much sequencing depth that corresponds to. This is particularly important for barcode assignment to phenotype in the long-read sequencing. In addition, a synonymous mutation comparison is mentioned but in my reading that data is not presented in the supplemental figures section.

We thank the reviewer for this succinct summary of the manuscript and the results. We appreciate the reviewer identifying data of interest that were not included in the original manuscript. We agree that this information is necessary to consider the results. Specific changes are summarized in the comments below.

The paper is very much written from an "old school" allostery perspective with static end point structures that are mutually exclusive - eg. p5110 "relative ligand-binding affinity between the two conformations" - however, an ensemble of conformations is likely needed to explain their data. This is especially true for the bandpass and inverted phenotypes they observe. The work by Hilser et al is of particular importance in this area. We would invite the authors to speculate more freely about the molecular origins of their findings.

We agree with the reviewer's suggestion, and we have adopted language to align more with the ensemble model of allostery. We continue to frame results using the Monad-Wyman-Changeux model,

which has been recently extended to predict the dose-response behavior of LacI and its dependence on a set of defined biophysical parameters. These models provide a framework in which to speculate on the biophysical origins of our results, and additional text has been added throughout the manuscript to discuss the possible biomolecular mechanics that underly the observed changes.

****Minor****

There are a number of small modifications. In general this paper is very technical and could use with some explanation and discussion for relevance to make the manuscript more approachable for a broader audience.

P1L23: Ligand binding at one site causes a conformational change that affects the activity of another > not necessarily true - and related to using more "modern" statistical mechanical language for describing allostery.

We agree with the reviewer's comment. We have addressed this comment by adopting language in line with more modern view of allostery, for example:

“With allosteric regulation, ligand binding at one site on a biomolecule changes the activity of another, often distal, site. Switching between active and inactive states provides a sense-and-response function that defines the allosteric phenotype.”

P2L20: The core experiment of this paper is a selection using a mutational library. In the main body the authors mention the library was created using mutagenic pcr but leave it at that. More details on what sort of mutagenic pcr was used in the main body would be useful. According to the methods error prone pcr was used. Why use er-pcr vs deep point mutational libraries? Presumably to sample higher order phenotype? Rationale should be included. Were there preliminary experiments that helped calibrate the mutation level?

We agree that justifying the decision to use error-prone PCR for library construction would be helpful. To explain this decision, we have added to the main text to explain this decision and to reflect on the consequences.

“We used error-prone PCR across the full lacI CDS to investigate the effects of higher-order substitutions spread across the entire LacI sequence and structure.”

And

“Novel phenotypes emerged at mutational distances greater than one amino acid substitution, highlighting the value in sampling a broader genotype space with higher-order mutations. Furthermore, the untargeted, random mutagenesis approach used here was critical for finding these novel phenotypes, as the genotypes required for these novel phenotypes were unpredictable.”

P2L20: Baseline library statistics would be great in a table for coverage, diversity, etc especially as this was done by error prone pcr vs a more saturated library generation method. This is present in sup fig2 but it's a bit complicated.

To more clearly convey the diversity within the library, we have included a heatmap of amino acid substitution counts found within the library (Appendix Fig. S4). Additionally, we have added Appendix Table S1, which lists the distribution of mutational distances of LacI variants found within the library, and the corresponding coverage of all possible mutations for each mutational distance.

P2L26: How were FACS gates drawn? This is in support fig17 - should be pointed to here.

We agree that a better description of the FACS process would be helpful. We have included Appendix Fig. S2, showing flow cytometry measurements of the library before and after FACS. Additionally, we have extended the description of the FACS process:

“The initial library had a bimodal distribution of G0, as indicated by flow cytometry results, with a mode at low fluorescence (near G0 of wild-type LacI), and a mode at higher gene expression. To generate a library in which most of the LacI variants could function as allosteric repressors, we used fluorescence activated cell sorting (FACS) to select the portion of the library with low fluorescence in the absence of ligand, gating at the trough between the two modes (Sony SH800S Cell Sorter, Appendix Fig S2).”

P3L4: Where is the figure/data for the synonymous SNP mutations? This should be in the supplement.

We agree this data is necessary to support the claim that LacI function was not impacted by synonymous mutations. We have included a new figure (Appendix Fig. S9) that shows the distribution of Hill equation parameters for LacI variants that code for the wild-type amino acid sequence, but with non-identical coding DNA sequences. Additionally, we included the results of a statistical analysis in the main text, this analysis compared all synonymous sequences in the library:

“We compared the distributions of the resulting Hill equation parameters between two sets of variants: 39 variants with exactly the wild-type coding DNA sequence for LacI (but with different DNA barcodes) and 310 variants with synonymous nucleotide changes (i.e. the wild-type amino acid sequence, but a non-wild-type DNA coding sequence). Using the Kolmogorov-Smirnov test, we found no significant differences between the two sets (p-values of 0.71, 0.40, 0.28, and 0.17 for G_0 , G_{∞} , EC_{50} , and n respectively, Appendix Fig. S9).”

P3L20: The authors use a ML learning deep neural network to predict variant that were not covered in the screen. However, the library generation method is using error prone pcr meaning there could multiple mutations resulting in the same amino acid change. The models performance was determined by looking at withheld data however error prone pcr could result in multiple nonsynonymous mutations of the same amino acid. For testing were mutations

truly withheld or was there overlap? Because several mutations are being represented by different codon combinations. Was the withheld data for the machine learning withholding specific substitutions?

We thank the reviewer for identifying the need to clarify this critical data analysis. Data was held-out at the amino acid level, and so no overlap between the training and testing datasets occurred. We have clarified the description of the method in the main text:

“We calculated RMSE using only held-out data not used in the model training, and the split between held-out data and training data was chosen so that all variants with a specific amino acid sequence appear in only one of the two sets.”

In addition, higher order protein interactions are complicated and idiosyncratic. I am surprised how well the neural net performs on higher order substitutions.

P4L4: Authors find mutations at the dimer/tetramer interfaces but don't mention whether polymerization is required. is dimerization required for dna binding? Tetramerization?

We agree with the reviewer that, overall, a description of LacI structure and function would improve the context for the reported results. As such, we have added Appendix Table S2, which defines the structural features discussed throughout the manuscript. Additionally, we strived to describe the relevant structural and functional role of specific amino acids that are discussed in the text. Finally, we have added a paragraph to the main text that summarizes the structure and function of LacI.

“The LacI protein has 360 amino acids arranged into three structural domains^{22–24}. The first 62 N-terminal amino acids form the DNA-binding domain, comprising a helix-turn-helix DNA-binding motif and a hinge that connects the DNA-binding motif and the core domain. The core domain, comprising amino acid positions 63–324, is divided into two structural subdomains: the N-terminal core and the C-terminal core. The full core domain forms the ligand-binding pocket, core-pivot region, and dimer interface. The tetramerization domain comprises the final 30 amino acids and includes a flexible linker and an 18 amino acid α -helix (Fig 3, Appendix Table S2). Naturally, LacI functions as a dimer of dimers: Two LacI monomers form a symmetric dimer that further assembles into a tetramer (a dimer of dimers).”

P4L8: Substitutions near the dimer interface both impact g_0 and ec_{50} , which authors say is consistent with a change in the allosteric constant. Can authors explain their thinking more in the paper to make it easier to follow? Are there any mutations in this area that only impact g_0 or ec_{50} alone? Why may these specific residues modify dimerization?

We agree that a more in-depth discussion on the possible mechanisms behind these phenotypic changes would improve the manuscript. We have added text throughout the subsection “Effects of amino acid substitutions on LacI phenotype,” we believe this added discussion improves the manuscript and clarifies the relationship between the observed allosteric phenotypes and the possible molecular mechanisms behind them.

P4L8: The authors discuss the allosteric constant extensively within the paper but do not explain it. It would be helpful to have an explanation of this to improve readability. This explanation should include the statistical mechanical basis of it and some speculation about the ways it manifests biophysically.

The allosteric constant is a critical concept, and we agree that it must be defined and discussed clearly throughout the manuscript. We have greatly expanded the discussion of the effects of single amino acid substitutions, and in the process, we give examples of biochemical changes in the protein, and how they may affect the allosteric constant. We think this added text improves the manuscript and helps clarify the allosteric constant and the biomolecular processes that affect it.

P4L1-16: Authors see mutations in the dimerization region that impact either G_0 and $G_{\text{saturated}}$ in combination with EC_{50} but not g_0 and $g_{\text{saturated}}$ together. Maybe we do not fully understand the hill equation but why are there no mutations that impact both g_0 and $g_{\text{saturated}}$ seen in support fig 13c? Why would mutations in the same region potentially impacting dimerization impact either g_0 or $g_{\text{saturated}}$? What might be the mechanism behind divergent responses?

It is important to recognize that the dimer interface does not just support the formation of dimers. There are many points of contact along the dimer interface that change when LacI switches between the active and inactive states. So, the dimer interface also helps regulate the balance between the active and inactive states. Our results show that different substitutions near the dimer interface can push this balance either toward the active or inactive states to varying degrees. We've added text throughout the description of single-substitutions effects to give specific examples and added a new paragraph at the end of that section to provide additional discussion and context. With regard to the more specific question of changes to both G_0 and G_{∞} , the biophysical models indicate that simultaneous changes to those Hill Equation parameters requires an unusual combination of biophysical changes. To clarify this point, we added a short paragraph to the text:

“None of the single amino substitutions measured in the library simultaneously decrease G_{∞} and increase G_0 (Appendix Fig S20C). This is not surprising, since substitutions that shift the biophysics to favor the active state tend to decrease G_{∞} while those that favor the inactive state tend to increase G_0 , and the biophysical models^{2,14,15} indicate that only a combination of parameter changes can cause both modifications to the dose-response. The library did, however, contain several multi-substitution variants with simultaneously decrease G_{∞} and increase G_0 . These inverted variants, and their associated substitutions are discussed below.”

P4L29: for interpretability it would be good to explain what log-additive effect means in the context of allostery.

We agree that this information would be useful to the reader and have added additional text to explain log-additivity. We thank the reviewer for pointing out this oversight.

“Combining multiple substitutions in a single protein almost always has a log-additive effect on EC_{50} . That is, the proportional effects of two individual amino acid substitutions on the EC_{50} can be multiplied together. For example, if

substitution A results in a 3-fold change, and substitution B results in a 2-fold change, the double substitution, AB, behaving log-additively, results in a 6-fold change.”

P4L34-P5L19: This section is wonderful. Really cool results and interesting structural overlap!

P5L34 Helix 9 of the protein is mentioned but it's functional relevance is not. This is common throughout the paper - it would be useful for there to be an overview somewhere to help the reader contextualize the results with known structural role of these elements.

We agree with the reviewer that this information would help to contextualize the results. We have made a number of changes to address this. First, we have added Appendix Table S2, which describes the structural features of LacI. Second, we have added a paragraph with a brief overview of the structure and function of LacI (discussed above). Third, we have expanded the section “the effects of individual amino acid substitutions on the function of LacI” to discuss the structural and/or biochemical impact of specific substitutions.

P5L39: The authors identified a triple mutant with the band-stop phenotype then made all combination of the triple mutant. Of particular interest is R195H/G265D which is nearly the same as the triple mutant. It would be nice if the positions of each of these mutations and have some discussion to begin to rationalize this phenotype, even if to point out how far apart they are and that there is no easy structural rationale!

We appreciate the reviewer highlighting this area of interest. We have added structural information to Fig. 6, which indicates to position of the amino acid substitutions that result in the band-stop phenotype, as well as a small discussion in the main text:

“To further investigate the band-stop phenotype, we chose a strong band-stop LacI variant with only three amino acid substitutions (R195H/G265D/A337D). These three positions are distributed distally on the periphery of the C-terminal core domain, and the role that each of these substitutions plays in the emergence of the band-stop phenotype is unclear.”

P6L9: There should be more discussion of the significance of this work directly compared to what is known. For instance, negative cooperativity is mentioned as an explanation for bi-phasic dose response but this idea is not explained. Why would the relevant free energy changes be more entropic? Another example is the reverse-TetR phenotype observed by Hillen et al.

We agree that more discussion is necessary to frame the results reported in the manuscript. To address this, we have added additional discussion throughout the manuscript that relates the results to the current understanding of allostery. Also, in the Discussion, we have added specific examples that lead us to link the ideas of bi-phasic dose response, negative cooperativity, and entropy/disorder. We believe these additions have improved the manuscript and we thank the reviewer for this suggestion.

P6L28: The authors mention that phenotypes exist with genotypes that are discoverable with genotype-

phenotype landscapes. This study due to the constraints of error prone pcr were somewhat limited. How big is the phenotypic landscape? Is it worth doing a more systematic study? What is the optimal experimental design: Single mutations, doubles, random - where is there the most information. How far can you drift before your machine learning model breaks down? How robust would it be to indels?

The reviewer raises some excellent questions here, some of which are appropriate subjects for future work. The optimal experimental design depends on the objective: If the goal is to understand every possible single mutation, a systematic site-saturation approach would be more appropriate. However, the landscape of a natural protein is limited by its wild-type DNA coding sequence, and so some substitutions are inaccessible (due to the arrangement of the codon table). The approach we took allowed to us characterize most of the accessible amino acid substitutions, while also allowing us to identify novel functions that would not have been identified with other approaches. We have added a little to the main text to discuss this (below). With regard to the DNN model, in the manuscript (Appendix Fig. S14), we show how the predictive accuracy degrades with mutational distance from the wild-type. It is possible that the type of DNN that we used could handle indels, since it effectively encodes each variant as a set of step-wise changes from the wild-type. But as with all machine-learning methods, it would require training with a dataset that included indels.

“Novel phenotypes emerged at mutational distances greater than one amino acid substitution, highlighting the value in sampling a broader genotype space with higher-order mutations. Furthermore, the untargeted, random mutagenesis approach used here was critical for finding these novel phenotypes, as the genotypes required for these novel phenotypes were unpredictable.”

Figures:

Sup figs 3-7: The comparison of library-based results and single mutants is a great example of how to validate genotype-phenotype experiments!

Thank you.

Supp fig 5.: Missing figure number.

We appreciate the reviewer catching this error and have attempted to properly label all figures and tables in this revision. Thank you.

Supp fig7: G0 appears to have very poor fit between library vs single mutant version. Why might this be? R² would likely be better to report here as opposed to RMSE as RMSE is sensitized to the magnitude of the data such that you cannot directly compare RMSE of say 'n' to G0.

We agree that these are important discussion points and have addressed this concern with an expanded discussion in the main text, as well as the addition of coefficient of correlation (R²) in the caption for Figure 2 (previously supplementary figure 7). We believe these additions contribute meaningfully to the manuscript, and they address the concerns of the reviewer. The additional text reads:

“We compared the Hill equation parameters from the library-scale measurement to those same parameters determined from flow cytometry measurements for each of the chemically synthesized LacI variants (Fig. 2). This served as a check of the new library-scale method’s overall ability to measure dose-response curves with quantitative accuracy. The accuracy for each Hill equation parameter in the library-scale measurement was: 4-fold for G_0 , 1.5-fold for G_∞ , 1.8-fold for EC_{50} , and ± 0.28 for n . For G_0 , G_∞ , and EC_{50} , we calculated the accuracy as: $\exp\left[\frac{\text{RMSE}(\ln(x))}{x}\right]$, where $\text{RMSE}(\ln(x))$ is the root-mean-square difference between the logarithm of each parameter from the library-scale and cytometry measurements. For n , we calculated the accuracy simply as the root-mean-square difference between the library-scale and cytometry results. The accuracy for the gene expression levels (G_0 and G_∞) was better at higher gene expression levels (typical for G_∞) than at low gene expression levels (typical for G_0), which is expected based on the non-linearity of the fitness impact of tetracycline (Appendix Figs. S10-S11). Measurements of the Hill coefficient, n , had high relative uncertainties for both barcode-sequencing and flow cytometry, and so the parameter n was not used in any quantitative analysis.”

Sup fig13c: it is somewhat surprising that mutations only appear to effect G_0 and not G_∞ . This implies that basal and saturated activity are not coupled. Is this expected? Why or why not?

This comment is partially addressed with a response above (P4L1-16). Beyond that, coupled gene expression increases do occur, notably with substitutions at the start codon that result in fewer copies of LacI in the cell. In this instance, both G_0 and G_∞ are increased. Other changes to the biophysical parameters could result in coupled changes to gene expression. Beyond that, measured changes to G_0 are less accurate, (discussed in the previous caption), and make finding smaller coupled effects difficult.

Reviewer #1 (Significance (Required)):

Allostery is hard to comprehend because it involves many interacting residues propagating information across a protein. The Monod-Wyman-Changeux (MWC) and Koshland, Nemethy, and Filmer (KNF) models have been a long standing framework to explain much of allostery, however recent formulations have focused on the role of the conformational ensemble and a grounding in statistical mechanics. This manuscript focuses on the functional impact of mutations and therefore contribution of the amino acids to regulation. The authors unbiased approach of combining a dose-response curve and mutational library generation let them fit every mutant to a hill equation. This approach let the authors identify the allosteric phenotype of all measured mutations! The authors found inverted phenotypes which happen in homologs of this protein but most interesting is the strange and idiosyncratic 'Band-stop' phenotype. The band-stop phenotype is bi-phasic that will hopefully be followed up with further studies to explain the mechanism. This manuscript is a fascinating exploration of the adaptability of allosteric landscapes with just a handful of mutations.

Genotype-phenotype experiments allow sampling immense mutational space to study complex phenotypes such as allostery. However, a challenge with these experiments is that allostery and other complicated phenomena come from immense fitness landscapes altering different parameters of the hill equation. The authors approach of using a simple error prone pcr library combined with many ligand concentrations allowed them to sample a very large space somewhat sparsely. However, they were able to predict this data by training and using a neural net model. I think this is a clever way to fill in the gaps that are inherent to somewhat sparse sampling from error prone pcr. The experimental design of the dose response is especially elegant and a great model for how to do these experiments.

With some small improvements for readability, this manuscript will surely find broad interest to the genotype-phenotype, protein science, allostery, structural biology, and biophysics fields.

We were prompted to do this by Review Commons and are posting our submitted review here:

Willow Coyote-Maestas has relevant expertise in high throughput screening, protein engineering, genotype-phenotype experiments, protein allostery, dating mining, and machine learning.

James Fraser has expertise in structural biology, genotype-phenotype experiments, protein allostery, protein dynamics, protein evolution, etc.

Reviewer #2 (Evidence, reproducibility and clarity (Required)):

The authors use deep mutational scanning to infer the dose-response curves of ~60,000 variants of the LacI repressor and so provide an unprecedentedly systematic dataset of how mutations affect an allosteric protein. Overall this is an interesting dataset that highlights the potential of mutational scanning for rapidly identifying diverse variants of proteins with desired or unexpected activities for synthetic biology/bioengineering. The relatively common inverted phenotypes and their sequence diversity is interesting, as is the identification of several hundred genotypes with non-sigmoidal band-stop dose-response curves and their enrichment in specific protein regions. A weakness of the study is that some of the parameter estimates seem to have high uncertainty and this is not clearly presented or the impact on the conclusions analysed. A second shortcoming is that there is little mechanistic insight beyond the enrichments of mutations with different effects in different regions of the protein. But as a first overview of the diversity of mutational effects on the dose-response curve of an allosteric protein, this is an important dataset and analysis.

****Comments****

****_Data quality and reproducibility_****

"The flow cytometry results confirmed both the qualitative and quantitative accuracy of the new method (Supplementary Figs. 3-7)"

- There need to be quantitative measures of accuracy in the text here for the different parameters.

We believe this comment is addressed along with the following two comments.

- Sup fig 7 panels should be main text panels - they are vital for understanding the data quality In particular, the GO parameter estimates from the library appear to have a lower bound ie

they provide no information below a cytometry G_0 of $\sim 10^4$. This is an important caveat and needs to be highlighted in the main text. The Hill parameter (n) estimate for wt (dark gray) replicate barcodes is extremely variable - why is this?

- In general there is not a clear enough presentation of the uncertainty and biases in the parameter estimations which seem to be rather different for the 4 parameters. Only the EC_{50} parameter seems to correlate very well with the independent measurements.

We thank the reviewer for identifying a need for more information on the accuracy of this method. So, we have moved Supplementary Fig. 7 to the main text (Fig 2 in the revised manuscript) and have added coefficients of correlation (R^2) to each Hill equation parameter in that figure caption. Furthermore, we have added new data (Appendix Fig. S11), which presents the uncertainty as a function of gene expression. Finally, we have added a discussion on the accuracy of this method for each parameter of the Hill equation to the main text. Estimation of the Hill coefficient (n) from data is often highly uncertain and variable, because that parameter estimate can be highly sensitive to random measurement errors at a single point on the curve. The estimate for the wild type appears to be highly variable because the plot contains 53 replicate measurements. So, the plotted variability represents approximately 2 standard deviations. The spread of wild-type results in the plot is consistent with the stated RMSE for the Hill coefficient. Furthermore, the Hill coefficient is not used in any of the additional quantitative analysis in our manuscript, partially because of its relatively high measurement uncertainty, but also because, based on the biophysical models, it is not as informative of the underlying biophysical changes.

“We compared the Hill equation parameters from the library-scale measurement to those same parameters determined from flow cytometry measurements for each of the chemically synthesized LacI variants (Fig. 2). This served as a check of the new library-scale method’s overall ability to measure dose-response curves with quantitative accuracy. The accuracy for each Hill equation parameter in the library-scale measurement was: 4-fold for G_0 , 1.5-fold for G_∞ , 1.8-fold for EC_{50} , and ± 0.28 for n . For G_0 , G_∞ , and EC_{50} , we calculated the accuracy as: $\exp\left[\frac{RMSE}{x}\right]$, where $RMSE$ ($\ln(x)$) is the root-mean-square difference between the logarithm of each parameter from the library-scale and cytometry measurements. For n , we calculated the accuracy simply as the root-mean-square difference between the library-scale and cytometry results. The accuracy for the gene expression levels (G_0 and G_∞) was better at higher gene expression levels (typical for G_∞) than at low gene expression levels (typical for G_0), which is expected based on the non-linearity of the fitness impact of tetracycline (Appendix Figs. S10-S11). Measurements of the Hill coefficient, n , had high relative uncertainties for both barcode-sequencing and flow cytometry, and so the parameter n was not used in any quantitative analysis.”

- The genotypes in the mutagenesis library contain a mean of 4.4 aa substitutions and the authors use a neural network to estimate 3 of the Hill equation parameters (with uncertainties) for the 1991/2110 of the single aa mutations. It would be useful to have an independent experimental evaluation of the reliability of these inferred single aa mutational effects by performing facs on a panel of single aa mutants (using single aa mutants in sup fig 3-7, if there are any, or newly constructed mutants).

We agree that the predictive performance of the DNN requires experimental validation. We evaluated the performance by withholding data from 20% of the library, including nearly 200 variants with single amino acid substitutions, and then compared the predicted effect of those substitutions to the measured effect. The results of this test are reported in Appendix Fig. S14. Additionally, we have adjusted the main text to more clearly explain the evaluation process.

“To evaluate the accuracy of the model predictions, we used the root-mean-square error (RMSE) for the model predictions compared with the measurement results. We calculated RMSE using only held-out data not used in the model training, and the split between held-out data and training data was chosen so that all variants with a specific amino acid sequence appear in only one of the two sets.”

- fig3/"Combining multiple substitutions in a single protein almost always has a log-additive effect on EC₅₀." How additive are the other 2 parameters? this analysis should also be presented in fig 3. If they are not as additive is it simply because of lower accuracy of the measurements? If the mutational effects are largely additive, then a simple linear model (rather than the DNN) could be used to estimate the single mutant effects from the multiple mutant genotypes.

We agree with the reviewer that exploring the log-additivity of the other Hill equation parameters is informative, and have included Appendix Figure S21, which displays this information. Furthermore, we expanded the discussion of log-additivity on all three parameters in the main text:

“Combining multiple substitutions in a single protein almost always has a log-additive effect on EC₅₀. That is, the proportional effects of two individual amino acid substitutions on the EC₅₀ can be multiplied together. For example, if substitution A results in a 3-fold change, and substitution B results in a 2-fold change, the double substitution, AB, behaving log-additively, results in a 6-fold change. Only 0.57% (12 of 2101) of double amino acid substitutions in the measured data have EC₅₀ values that differ from the log-additive effects of the single substitutions by more than 2.5-fold (Fig 4). This result, combined with the wide distribution of residues that affect EC₅₀, reinforces the view that allostery is a distributed biophysical phenomenon controlled by a free energy balance with additive contributions from many residues and interactions, a mechanism proposed previously^{1,36} and supported by other recent studies¹⁷, rather than a process driven by the propagation of local, contiguous structural rearrangements along a defined pathway.

A similar analysis of log-additivity for G₀ and G_∞ is complicated by the more limited range of measured values for those parameters, the smaller number of substitutions that cause large shifts in G₀ or G_∞, and the higher relative measurement uncertainty at low G(L). However, the effects of multiple substitutions on G₀ and G_∞ are also consistent with log-additivity for almost every measured double substitution variant (Appendix Fig S21).”

Also, we agree that the DNN model should be compared to a simple linear-additive model. So, we have added a comparison of prediction performance for the recurrent DNN model used in the manuscript, a linear-additive model, and a feed-forward DNN model (Appendix Fig. S12). *We also discuss the evaluation of the three models in the main text:*

“We tested two different neural network architectures: a recurrent DNN and a more conventional feed-forward DNN, as well as a linear-additive model. Of the three models, the recurrent DNN model provides the best predictive performance for each of the Hill equation parameters, though for EC_{50} , the recurrent DNN and linear-additive models have similar performance (Appendix Fig. S12). So, for subsequent analysis, we used the recurrent DNN model, which captures the context dependence of amino acid substitution effects (Appendix Fig S12).”

*****_Presentation/clarity of text and figures_*****

- The main text implies that the DNN is trained to predict 3 parameters of the Hill equation but not the Hill coefficient (n). This should be clarified / justified in the main text.

We agree that the decision to exclude the parameter ‘ n ’ requires explanation in the main text. To address this, we have added to the main text:

“Measurements of the Hill coefficient, n , had high relative uncertainties for both barcode-sequencing and flow cytometry, and so the parameter n was not used in any quantitative analysis.”

and

“We trained the model to predict the Hill equation parameters G_0 , G_{∞} , and EC_{50} (Appendix Fig. S13), the three Hill equation parameters that were determined with relatively low uncertainty by the library-scale measurement.”

- The DNN needs to be better explained and justified in the main text for a general audience. How do simpler additive models perform for phenotypic prediction / parameter inference?

We agree that the DNN model should be compared to other predictive models. We have added a comparison of prediction performance for the recurrent DNN model used in the manuscript, a linear-additive model, and a feed-forward DNN model (Appendix Fig. S12). We also discuss the evaluation of the three models in the main text.

“So, to comprehensively determine the impact of single amino acid substitutions, we constructed a deep neural network model (DNN) capable of accurately

predicting the Hill equation parameters for LacI variants that were not directly measured. We tested two different neural network architectures: a recurrent DNN and a more conventional feed-forward DNN, as well as a linear-additive model. Of the three models, the recurrent DNN model provides the best predictive performance for each of the Hill equation parameters, though for EC_{50} , the recurrent DNN and linear-additive models have similar performance (Appendix Fig. S12). So, for subsequent analysis, we used the recurrent DNN model, which captures the context dependence of amino acid substitution effects (Appendix Fig S12). In addition, to estimate uncertainties for the model predictions, we used approximate Bayesian inference methods as described in the Materials and Methods¹⁹.”

- Ref 14. analyses a much smaller set of mutants in the same protein but using an explicit biophysical model. It would be helpful to have a more extensive comparison with the approach and conclusions to this previous study.

We agree and have added text throughout the manuscript to frame the results and discussion in terms of the referenced biophysical models. Using the models, we describe the biophysical effects that a substitution may have on LacI, based on observed changes to function associated with that substitution. We also comment briefly on the limitations of these models when applied to the extensive dataset presented here.

“Most of the non-silent substitutions discussed above are more likely to affect the allosteric constant than either the ligand or operator affinities. Within the biophysical models, those affinities are specific to either the active or inactive state of LacI, i.e. they are defined conditionally, assuming that the protein is in the appropriate state. So, almost by definition, substitutions that affect the ligand-binding or operator-binding affinities (as defined in the models) must be at positions that are close to the ligand-binding site or within the DNA-binding domain. Substitutions that modify the ability of the LacI protein to access either the active state or inactive state, by definition, affect the allosteric constant. This includes, for example, substitutions that disrupt dimer formation (dissociated monomers are in the inactive state), substitutions that lock the dimer rigidly into either the active or inactive state, or substitutions that more subtly affect the balance between the active and inactive states. Thus, because there are many more positions far from the ligand- and DNA- binding regions than close to those regions, there are many more opportunities for substitutions to affect the allosteric constant than the other biophysical parameters. Note that this analysis assumes that substitutions don’t perturb the LacI structure too much, so that the active and inactive states remain somehow similar to the wild-type states. Our results suggest that this is not always the case: consider, for example, the substitutions at positions K84 and M98 discussed above and the substitutions resulting in the inverted and band-stop phenotypes discussed below.”

- Enrichments need statistical tests to know how unexpected that results are e.g. p5 line 12
"67% of strongly inverted variants have substitutions near the ligand-binding pocket"

We agree that this information is necessary to interpret the results. We have included p-values (previously reported only in the Methods section) throughout the main text of the manuscript.

- missing citation: Poelwijk et al 2011 [https://www.cell.com/fulltext/S0092-8674\(11\)00710-0](https://www.cell.com/fulltext/S0092-8674(11)00710-0)
previously reported an inverted dose-response curve for a *lacI* mutant.

The publication by Poelwijk *et al.* was considered extensively when planning this work, and failing to cite that work would have been tremendously unjust. We have included it, as well as a few additional references that have identified and discussed inverted *LacI* variants. We sincerely thank the reviewer for identifying this oversight.

- What mechanisms do the authors envisage that could produce the band-stop dose response curves? There is likely previous theoretical work that could be cited here. In general there is little discussion of the biophysical mechanisms that could underlie the various mutational effects.

We agree with the reviewer, that discussing the biophysical mechanisms that underlie many of the reported mutations is important to understand the results. We have expanded the subsection "Effects of amino acid substitutions on *LacI* phenotype" to include discussion on several of the key substitutions (or groups of substitutions) and their potential biophysical effects. Additionally, we consider mechanism that may underlie the band-stop sensor, and propose a possible mechanism to explain the band-stop phenotype:

"In particular, the biphasic dose response of the band-stop variants suggests negative cooperativity: that is, successive ligand binding steps have reduced ligand binding affinity. Negative cooperativity has been shown to be required for biphasic dose-response curves^{39,40}. The biphasic dose-response and apparent negative cooperativity are also reminiscent of systems where protein disorder and dynamics have been shown to play an important role in allosteric function¹, including catabolite activator protein (CAP)^{41,42} and the Doc/Phd toxin-antitoxin system⁴³. This suggests that entropic changes may also be important for the band-stop phenotype. A potential mechanism is that band-stop *LacI* variants have two distinct inactive states: an inactive monomeric state and an inactive dimeric state. In the absence of ligand, inactive monomers may dominate the population. Then, at intermediate ligand concentrations, ligand binding stabilizes dimerization of *LacI* into an active state which can bind to the DNA operator and repress transcription. When a second ligand binds to the dimer, it returns to an inactive dimeric state, similar to wild-type *LacI*. Similar dimerization-based regulation has been described before, and supports the observed negative cooperativity and biphasic dose-response³⁹. This mechanism, and other possible mechanisms, do not match the MWC model of allostery or its extensions^{2,13-15} and require a more comprehensive

study and understanding of the ensemble of states in which these band-stop LacI variants exist.”

- *"This result, combined with the wide distribution of residues that affect EC50, suggests that LacI allostery is controlled by a free energy balance with additive contributions from many residues and interactions." 'additive contributions and interactions' covers all possible models of vastly different complexity i.e. this sentence is rather meaningless.*

We have attempted to contextualize this statement by adding additional discussion and references. We hope these additions give more meaning to this section.

“This result, combined with the wide distribution of residues that affect EC₅₀, reinforces the view that allostery is a distributed biophysical phenomenon controlled by a free energy balance with additive contributions from many residues and interactions, a mechanism proposed previously^{1,39} and supported by other recent studies¹⁷, rather than a process driven by the propagation of local, contiguous structural rearrangements along a defined pathway.”

- *fig 4 c and d compress a lot of information into one figure and I found this figure confusing. It may be clearer to have multiple panels with each panel presenting one aspect. It is also not clear to me what the small circular nodes exactly represent, especially when you have one smaller node connected to two polygonal nodes, and why they don't have the same colour scale as the polygonal nodes.*

We agree with the reviewer that Figure 4 (or Figure 5 in the revised manuscript) contains a lot of information. The purpose of this figure is to convey the structural and genetic diversity among the sets of inverted variants and band-stop variants. We designed this figure to convey this point at two levels: a brief overview, where the diversity is apparent by quickly considering the figure, and at a more informative level, with some quantitative data and structurally relevant points highlighted. We have modified the caption slightly, in an effort to improve clarity. Additionally, the source data used to generate this plot is included with the manuscript, to allow a reader further investigation into this data.

- *line 25 - 'causes a conformational change' -> 'energetic change' (allostery does not always involve conformational change*

We thank the reviewer for this comment and have adopted a more modern language describe allostery throughout the manuscript.

- *sup fig 5 legend misses '5'*

We thank the reviewer for pointing this out, we have attempted to number all figures and tables more carefully.

- sup fig 7. pls add correlation coefficients to these plots (and move to main text figures).

We agree that this information is of interest and have included this data as main text Figure 2. In addition, we have included coefficients of correlation in the caption of this figure.

- Reference 21 is just a title and pubmed link

We thank the reviewer for identifying this error, we have corrected this in the references.

- "fitness per hour" -> growth rate

To ensure that this connection is clearly established, when we introduce fitness for the first time we clarify that it relates to growth rate:

“Consequently, in the presence of tetracycline, the LacI dose-response modulates cellular fitness (i.e. growth rate) based on the concentration of the input ligand isopropyl-β-D-thiogalactoside (IPTG).”

We also define ‘fitness’ in the Methods section:

“The experimental approach for this work was designed to maintain bacterial cultures in exponential growth phase for the full duration of the measurements. So, in all analysis, the Malthusian definition of fitness was used, i.e. fitness is the exponential growth rate⁵⁸.”

- page 6 line 28 - "discoverable only via large-scale landscape measurements" - directed evolution approaches can also discover such genotypes (see e.g. Poelwijk /Tans paper). Please re-phrase.

We agree with the reviewer and have adjusted the main text accordingly.

“Overall, our findings suggest that a surprising diversity of useful and potentially novel allosteric phenotypes exist with genotypes that are readily discoverable via large-scale landscape measurements.”

- pls define jargon the first time it is used e.g. band-stop and band-pass

We agree that all unconventional terms should be explicitly defined when used, and we have attempted to define the band-pass and band-stop dose-response curves more clearly in the main text:

“These include examples of LacI variants with band-stop dose-response curves (i.e. variants with high-low-high gene expression; e.g. Fig. 1e, Appendix Fig. S7), and LacI variants with band-pass dose-response curves (i.e. variants with low-high-low gene expression; e.g. Appendix Fig. S8).”

****_Methods/data availability/ experimental and analysis reproducibility:_****

The way that growth rate is calculated on page 17 equation 1- This section is confusing. Please be explicit about how you accounted for the lag phase, what the lag phase was, and total population growth during this time. In addition, please report the growth curves from the wells of the four plates, the final OD600 of the pooled samples, and exact timings of when the samples were removed from 37 degree incubation in a table. These are critical for calculating growth rate in individual clones downstream.

We thank the reviewer for identifying the need to clarify this section of text and to provide relevant data. First, we have included Appendix Table S5, which includes the final OD600 values for every chemical growth condition. Second, the ‘lag’ in this section referred to a delay before tetracycline began impacting the growth rate of cells and not the ‘lag phase’ normally associated with bacterial growth measurements. To address this, we have changed ‘lag’ in this context to ‘delay.’ Furthermore, we have attempted to clarify precisely the cause of this delay, and how we accounted for it in calculating growth rates:

For samples grown with tetracycline, the tetracycline was only added to the culture media for Growth Plates 2-4. Because of the mode of action of tetracycline (inhibition of translation), there was a delay in its effect on cell fitness: Immediately after diluting cells into Growth Plate 2 (the first plate with tetracycline), the cells still had a normal level of proteins needed for growth and proliferation and they continued to grow at nearly the same rate as without tetracycline. Over time, as the level of proteins required for cell growth decreased due to tetracycline, the growth rate of the cells decreased. Accordingly, the analysis accounts for the variation in cell fitness (growth rate) as a function of time after the cells were exposed to tetracycline. With the assumption that the fitness is approximately proportional to the number of proteins needed for growth, the fitness as a function of time is taken to approach the new value with an exponential decay:

$$\mu_{i,j} = \mu_i^0 + (\mu_i^{\text{tet}} - \mu_i^0)e^{-\alpha j} \quad (3)$$

where μ_i^{tet} is the steady-state fitness with tetracycline, and α is a transition rate. The transition rate was kept fixed at $\alpha = \log(5)$, determined from a small-scale calibration measurement. Note that at the tetracycline concentration used during the library-scale measurement (20 $\mu\text{g}/\text{mL}$), μ_i^{tet} was greater than zero even at the lowest G(L) levels (Appendix Fig. S10). From Eq. (3), the number of cells in each Growth Plate for samples grown with tetracycline is:

- What were the upper and lower bounds of the measurements? (LacI deletion vs Tet deletion / autofluorescence phenotype - true 100% and true 0% activity). Knowing and reporting these bounds will also allow easier comparison between datasets in the future.

We agree that knowing the limitations of the measurement are important for contextualizing the results. To address this point, we have included Appendix Fig. S11, which shows the uncertainty of the measurement across gene expression levels. We also report all results in traceable, quantitative units (Molecules of equivalent fluorophore, MEF), to enable comparability with other work. We have also included information on the autofluorescence control measurements and their use in the data analysis.

“An autofluorescence control (strain MG1655 Δ lac with a plasmid similar to pVER but lacking the YFP gene) was also measured with flow cytometry and analyzed in the same way as other variants. The regulated gene expression output level for each variant is reported as the geometric mean of the measured fluorescence for that variant minus the geometric mean of the measured fluorescence for the zero-fluorescence control (92 MEF).”

Please clarify whether there was only 1 biological replicate (because the plates were pooled before sequencing)? Or if there were replicates present an analysis of reproducibility.

We thank the reviewer for pointing out the ambiguity in the original manuscript. The library-scale measurement reported here was completed once, the 24 growth conditions were spread across 96 wells, so each condition occupied 4 wells. The 4 wells were combined prior to DNA extraction. We have clarified this process in the methods by removing ‘duplicate’ and changing the text:

“Growth Plate 2 contained the same IPTG gradient as Growth Plate 1 with the addition of tetracycline (20 μ g/mL) to alternating rows in the plate, resulting in 24 chemical environments, with each environment spread across 4 wells.”

Despite there being only a single library-scale measurement, the accuracy and reliability of the results are supported by many distinct biological replicates within the library (i.e. LacI variants with the same amino acid sequence but with different barcodes, see new Appendix Fig. S9), as well as over 100 orthogonal dose-response curve measurements completed with flow cytometry (Figure 2). We believe these support the reproducibility of the work and we have included statistical analysis on the accuracy of the library-scale measurement results.

“To evaluate the accuracy of the new method for library-scale dose-response curve measurements, we independently verified the results for over 100 LacI variants from the library. For each verification measurement, we chemically synthesized the coding DNA sequence for a single variant and inserted it into a plasmid where LacI regulates the expression of a fluorescent protein. We transformed the plasmid into E. coli and measured the resulting dose-response curve with flow cytometry (e.g. Fig. 1e). We compared the Hill equation parameters from the library-scale measurement to those same parameters determined from flow cytometry measurements for each of the chemically synthesized LacI variants (Fig. 2). This

served as a check of the new library-scale method's overall ability to measure dose-response curves with quantitative accuracy. The accuracy for each Hill equation parameter in the library-scale measurement was: 4-fold for G_0 , 1.5-fold for G_∞ , 1.8-fold for EC_{50} , and ± 0.28 for n . For G_0 , G_∞ , and EC_{50} , we calculated the accuracy as: " \exp " ["RMSE" ("ln" ("x"))], where "RMSE" ("ln" ("x")) is the root-mean-square difference between the logarithm of each parameter from the library-scale and cytometry measurements. For n , we calculated the accuracy simply as the root-mean-square difference between the library-scale and cytometry results (Appendix Fig. S7)."

- Please provide supplementary tables of the data (in addition to the raw sequencing files). Both a table summarising the growth rates, inferred parameter values and uncertainties for genotypes and a second table with the barcode sequence counts across timepoints and associated experimental data.

We agree that access to this information is critical. Due to the size of the associated data, we have made this data available for download in a public repository. We direct readers to the repository information in the "Data Availability" statement:

*"The processed data table containing comprehensive data and information for each *lacI* variant in the library is publicly available via the NIST Science Data Portal, with the identifier [ark:/88434/mds2-2259](https://data.nist.gov/od/id/mds2-2259) (<https://data.nist.gov/od/id/mds2-2259> or <https://doi.org/10.18434/M32259>). The data table includes the DNA barcode sequences, the barcode read counts for each sample and time point used for the library-scale measurement, fitness estimates for each barcoded variant across the 24 chemical environments, the results of both Bayesian inference models (including posterior medians, covariances, and 0.05, 0.25, 0.75, and 0.95 posterior quantiles), the *lacI* CDS and amino acid sequence for each barcoded variant (as determined by long-read sequencing), the number of *lacI* CDS reads in the long-read sequencing dataset for each barcoded variant, and the number of unintended mutations in other regions of the plasmid (from the long-read sequencing data).*

All data analysis code is available at https://github.com/djross22/nist_lacI_landscape_analysis."

Reviewer #2 (Significance (Required)):

The authors present an unprecedentedly systematic dataset of how mutations affect an allosteric protein. This illustrates the potential of mutational scanning for rapidly identifying diverse variants of allosteric proteins / regulators with desired or unexpected activities for synthetic biology/bioengineering.

*Previous studies have identified inverted dose-response curve for a *lacI* phenotypes [https://www.cell.com/fulltext/S0092-8674\(11\)00710-0](https://www.cell.com/fulltext/S0092-8674(11)00710-0) but using directed evolution i.e. they were not comprehensive in nature.*

The audience of this study would be protein engineers, the allostery field, synthetic biologists and the mutation scanning community and evolutionary biologists interested in fitness landscapes.

My relevant expertise is in deep mutational scanning and genotype-phenotype landscapes, including work on allosteric proteins and computational methods.

Reviewer #3 (Evidence, reproducibility and clarity (Required)):

*In this interesting manuscript the authors developed an ingenious high throughput screening approach which utilizes DNA barcoding to select variants of LacI proteins with different allosteric profiles for IPTG control using *E. coli* fitness (growth rate) in a range of antibiotic concentrations as a readout thus providing a genotype-phenotype map for this enzyme. The authors used a library of 10^5 - 10^6 variants of LacI expressed from a plasmid and screened for distinct IPTG activation profiles under different conditions including several antibiotic stressors. As a result they identified various patterns of activation including normal (sigmoidal increase), inverted (decrease) and unusual stop-band where the dependence of growth on [IPTG] is non-monotonic. The study is well-conceived, well executed and provides statistically significant results. The key advance provided by this work is that it allows to identify specific mutations in LacI connected with one of three allosteric profiles. The paper is clearly written all protocols are explained and it can be reproduced in a lab that possesses proper expertise in genetics.*

Reviewer #3 (Significance (Required)):

The significance of this work is that it discovered libraries of LacI variants which give rise to distinct profiles of allosteric control of activation of specific genes (in this case antibiotic resistance) by the Lac mechanism. The barcoding technology allowed to identify specific mutations which are (presumably) causal of changes in the way how allosteric activation of LacI by IPTG works. As such it provides a rich highly resolved dataset of LacI variants for further exploration and analysis.

Alongside with these strengths several weaknesses should also be noted:

1. First and foremost the paper does not provide any molecular-level biophysical insights into the impact of various types of mutations on molecular properties of LacI. Do the mutations change binding affinity to IPTG? Binding site? Communication dynamics? Stability? The diagrams of connectivity for the stop-band mutations (Fig.4) do not provide much help as they do not tell much which molecular properties of LacI are affected by mutations and why certain mutations have specific effect on allostery. A molecular level exploration would make this paper much stronger.

We address this comment with comment (2), below.

2. In the same vein a theoretical MD study would be quite illuminating in answering the key unanswered question of this work: Why do mutations have various and pronounced effects of allosteric regulation by LacI?. I think publication of this work should not be conditioned on such study but again adding would make the work much stronger.

We appreciate the reviewer's comments and agree that investigating the molecular mechanisms driving the phenotypic changes identified in this work is a compelling proposition. Throughout the manuscript, we identify positions and specific amino acid substitutions that affect the measurable function of LacI. We also discuss the biophysical effects that may underly these changes. In this revision, we have expanded the discussion to speculate on possible molecular-level effects. However, examining these

effects, either computationally or experimentally, will require additional work that we think is beyond the scope of this manuscript, although we hope that such studies do occur.

3. Lastly a recent study PNAS v.116 pp.11265-74 (2019) explored a library of variants of E. coli Adenylate Kinase and showed the relationship between allosteric effects due to substrate inhibition and stability of the protein. Perhaps a similar relationship can explored in this case of LacI.

We thank the reviewer for highlighting this publication. We agree with the reviewer that similar effects may play a role in the activity of LacI. Establishing such relationships would certainly improve our understanding of biology. We hope the method and dataset reported here will spur investigations of this phenomenon and other related mechanisms that may underlie the band-stop phenotype and other observed effects.

Thank you again for submitting your work to Molecular Systems Biology along with the referee reports from Review Commons. We have now heard back from reviewer #2 who was asked to evaluate your revised study. As you will see below, the reviewer thinks that their concerns have been satisfactorily addressed and is supportive of publication in Molecular Systems Biology.

Before we can formally accept the study for publication, we would ask you to address the following editorial issues.

REFeree REPORTS

Reviewer #2:

The authors have addressed my concerns. I think this is an extremely interesting dataset and analysis that provides the first comprehensive analysis of how mutations affect the dose-response function of an allosteric protein.

The authors have made all requested editorial changes.

2nd Revision - Editorial Decision

18th Feb 2021

Thank you again for sending us your revised manuscript. We are now satisfied with the modifications made and I am pleased to inform you that your paper has been accepted for publication.

Corresponding Author Name: David Ross

Manuscript Number: MSB-2020-10179